# Comparison of rhesus and cynomolgus macaques as an infection model for COVID-19

Francisco J. Salguero[1], Andrew D. White [1], Gillian S. Slack[1], Susan A. Fotheringham[1], Kevin R. Bewley [1], Karen E. Gooch[1], Stephanie Longet [1], Holly E. Humphries[1], Robert J. Watson[1], Laura Hunter[1], Kathryn A. Ryan [1], Yper Hall[1], Laura Sibley[1], Charlotte Sarfas[1], Lauren Allen[1], Marilyn Aram[1], Emily Brunt[1], Phillip Brown [1], Karen R. Buttigieg[1], Breeze E. Cavell [1], Rebecca Cobb[1], Naomi S. Coombes[1], Alistair Darby[2], Owen Daykin-Pont[1], Michael J. Elmore [1], Isabel Garcia-Dorival[2], Konstantinos Gkolfinos[1], Kerry J. Godwin[1], Jade Gouriet[1], Rachel Halkerston[1], Debbie J. Harris[1], Thomas Hender[1], Catherine M. K. Ho[1], Chelsea L. Kennard[1], Daniel Knott[1], Stephanie Leung[1], Vanessa Lucas[1], Adam Mabbutt[1], Alexandra L. Morrison[1], Charlotte Nelson[2], Didier Ngabo [1], Jemma Paterson[1], Elizabeth J. Penn[1], Steve Pullan[1], Irene Taylor[1], Tom Tipton [1], Stephen Thomas[1], Julia A. Tree[1], Carrie Turner[1], Edith Vamos[2], Nadina Wand [1], Nathan R. Wiblin[1], Sue Charlton[1], Xiaofeng Dong[2], Bassam Hallis[1], Geoffrey Pearson[1], Emma L. Rayner[1], Andrew G. Nicholson[3], Simon G. Funnell [1], Julian A. Hiscox[2,4], Mike J. Dennis[1], Fergus V. Gleeson[5], Sally Sharpe[1] & Miles W. Carroll[1,6✉]

A novel coronavirus, SARS-CoV-2, has been identified as the causative agent of the current COVID-19 pandemic. Animal models, and in particular non-human primates, are essential to understand the pathogenesis of emerging diseases and to assess the safety and efficacy of novel vaccines and therapeutics. Here, we show that SARS-CoV-2 replicates in the upper and lower respiratory tract and causes pulmonary lesions in both rhesus and cynomolgus macaques. Immune responses against SARS-CoV-2 are also similar in both species and equivalent to those reported in milder infections and convalescent human patients. This finding is reiterated by our transcriptional analysis of respiratory samples revealing the global response to infection. We describe a new method for lung histopathology scoring that will provide a metric to enable clearer decision making for this key endpoint. In contrast to prior publications, in which rhesus are accepted to be the preferred study species, we provide convincing evidence that both macaque species authentically represent mild to moderate forms of COVID-19 observed in the majority of the human population and both species should be used to evaluate the safety and efficacy of interventions against SARS-CoV-2. Importantly, accessing cynomolgus macaques will greatly alleviate the pressures on current rhesus stocks.

[1] National Infection Service, Public Health England (PHE), Porton Down, Salisbury, Wiltshire, UK. [2] Institute of Infection, Veterinary and Ecological Sciences, University of Liverpool, Liverpool, UK. [3] Royal Brompton and Harefield NHS Foundation Trust, and National Heart and Lung Institute, Imperial College, London, UK. [4] Infectious Diseases Horizontal Technology Centre (ID HTC), A*STAR, Singapore, Singapore. [5] Department of Oncology, Oxford University, Oxford, UK. [6] Nuffield Department of Medicine, Wellcome Trust Centre for Human Genetics, Oxford University, Oxford OX3 7BN, UK. ✉email: miles.carroll@phe.gov.uk

A novel acute respiratory syndrome, now called Coronavirus disease-19 (COVID-19) was first reported in China in December 2019. The genetic sequence of the causative agent was found to have similarity with two highly pathogenic respiratory beta Coronaviruses, SARS-CoV-1[1] and MERS[2], and was later called SARS-CoV-2[3]. It has currently infected >57 million individuals resulting in >1.3 million deaths[4]. Among the clinical and pathological signs of SARS-CoV-2 infection in humans, pneumonia accompanied by respiratory distress seem to be the most clinically relevant[5,6].

The development of animal models that replicate human disease is a crucial step in the study of pathogenesis and transmission, in addition to the assessment of the safety and efficacy of candidate vaccines and therapeutics. Due to their obvious physiological similarities to humans, non-human primates (NHPs), such as macaques, have long been recognised as the most clinically relevant animal for the development of in vivo models of human disease. Prior NHP models developed for SARS-CoV-1 and MERS have shown respiratory induced pathology with similar features as those seen in humans, including diffuse alveolar damage. For SARS-CoV-1, though rhesus macaques became the preferred species, yet the literature suggests there is no significant difference in susceptibility of cynomolgus (*Macaca fascicularis*) or rhesus macaques (*Macaca mulatta*) to infection, virus replication and pathology[7–10]. Recent studies have shown that rhesus macaques[11,12] and cynomolgus macaques[13] can be infected by SARS-CoV-2 and the disease course resembles some features of human COVID-19 infection. Rhesus macaques appear to display more extensive lung pathology and clinical signs[14]; however, a direct comparison of species, based on these studies, is not possible because different strains of virus, dose and route of administration have been used. A head-to-head comparison of rhesus versus cynomolgus macaques, under the same experimental conditions, is urgently required as there are now acute pressures on rhesus stocks that will impact on the ability to perform safety and efficacy studies on new vaccines and therapeutics.

This unique study aims to evaluate the outcome of exposure to SARS-CoV-2 in a head-to-head comparison of two macaque species: rhesus macaque (Indian genotype) and cynomolgus macaque (Mauritian genotype) exposed to the same well characterised strain of challenge virus.

Similar to recent NHP studies, a dose of $5 \times 10^6$ pfu of SARS-CoV-2/Victoria/01/2020[15], was delivered via the intranasal (IN) and intratracheal (IT) route to groups of each species comprising six animals. We perform sequential body fluid sampling and culls (at days 4/5, 14/15 and 18/19) to support a comprehensive comparative assessment of clinical signs, pathology, virology and immunology. The results indicate parallel URT and LRT virus replication, patterns of lung pathology and adaptive immunity. This comparative data is reiterated in the global transcriptomics response between the two species that closely resemble a mild to moderate COVID-19 in humans.

## Results

**Clinical signs and in-life imaging by CT scan.** Six rhesus macaques of Indian genotype and six cynomolgus macaques of Mauritian genotype were obtained from established UK Government breeding colonies (Fig. 1). Study groups comprised three males and three females of each species and all were adults aged two to four years with body weights ranging between 2.89 and 4.85 kg at time of challenge. Animals were challenged with a total of $5 \times 10^6$ pfu of SARS-CoV-2/Victoria/01/2020 administered in volumes of 2 ml by the intratracheal route (using a bronchoscope for accurate placement right above the carinal bifurcation) and 1 ml by the intranasal route (0.5 ml in each nostril). Whole-genome sequencing of the viral challenge stock, used in this study (Passage 3), confirmed there were no significant changes following passage in Vero/hSLAM cells, compared to the original isolate (Passage 1).

No significant weight loss or changes in body temperature were observed throughout the experiment. Adverse clinical signs were not recorded for any animal despite frequent monitoring during the study period.

Images from CT scans collected 18 days after challenge from two rhesus and two cynomolgus macaques, were examined by an expert thoracic radiologist with prior experience of non-human primate CT interpretation and human COVID-19 CT features, blinded to the clinical status. Pulmonary abnormalities that involved less than 25% of the lung and reflected those characteristics of SARS-CoV-2 infection in humans, were identified in one rhesus macaque and both cynomolgus macaques (Fig. 2). Ground glass opacity was observed in all three macaques showing abnormal lung structure, with

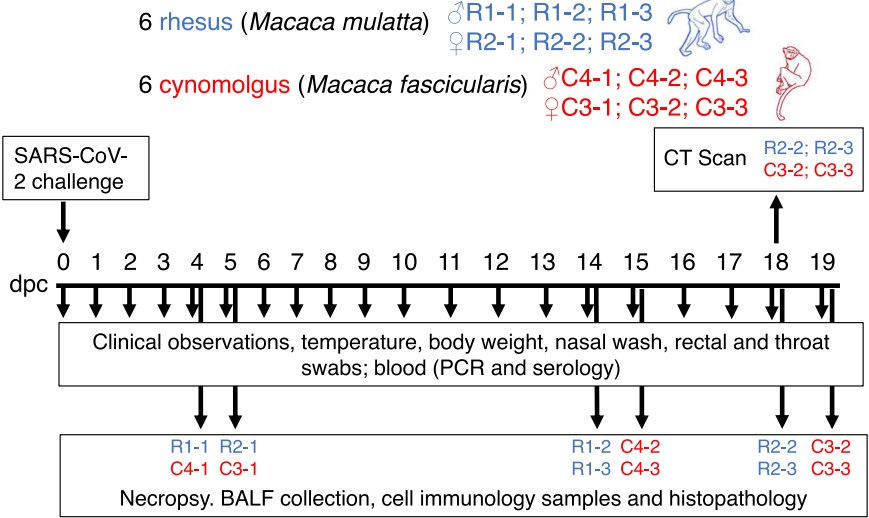

**Fig. 1 Experimental study design.** Rhesus and cynomolgus macaques were inoculated with severe acute respiratory syndrome coronavirus 2 (SARS-CoV-2) and culled in groups at three time points (4/5; 14/15 and 18/19 dpc). Animals were monitored for clinical signs and blood/swab/tissue samples were taken for virology, immunology and pathology. CT scan imaging was carried out at 18 dpc.

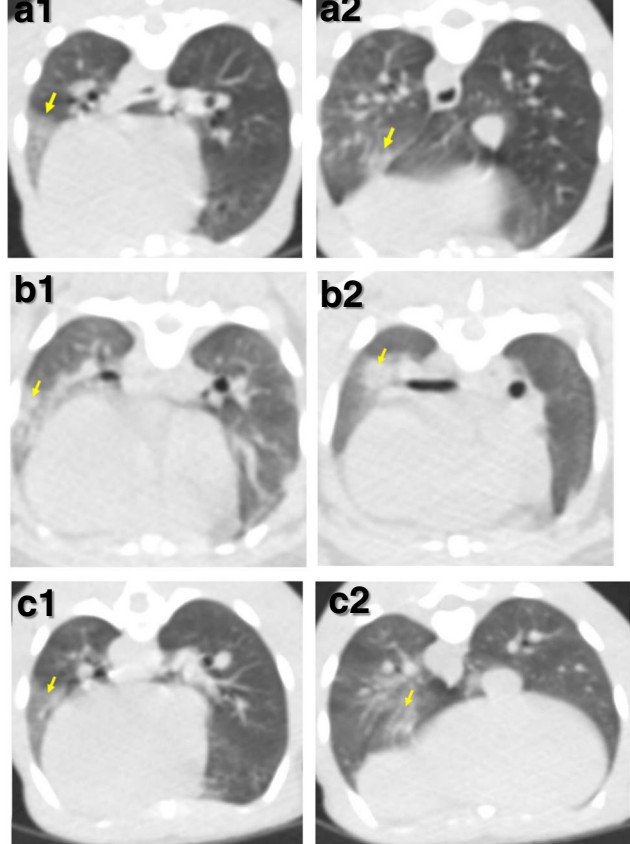

**Fig. 2 CTscan images from cynomolgus and rhesus macaques infected with SARS-CoV-2 and culled at 18 days post challenge.** Images constructed from CT scans collected 18 days after challenge with SARS-CoV-2 showing pulmonary abnormalities in two cynomolgus (**a**, **b**) and one rhesus macaque (**c**). Arrows in images (**a1**), (**b1**) and (**c1**) indicate areas of peripheral ground glass opacification. Arrows in images (**a2**) and (**c2**) indicate areas of ground glass opacification and arrow in image (**b2**) indicates an area of consolidation. Images from a second rhesus macaque did not have abnormal features.

peripheral consolidation also seen in one cynomolgus macaque. Abnormalities occurred in the peripheral two thirds of the middle and lower lung lobes in the two cynomolgus macaques and in a random pattern in the upper, middle and lower lobes of the rhesus macaque. Pulmonary emboli were not identified in any of the subjects.

**Viral load in clinical samples.** Viral load in the upper respiratory tract (URT) (nasal washes and throat swabs), gastrointestinal tract (rectal swabs) and in systemic samples (EDTA blood) was assessed by CDC N-gene RT-qPCR and subgenomic E-gene RT-qPCR at regular intervals throughout the study, and in bronchioalveolar lavage (BAL) collected at necropsy (Fig. 3). High levels of viral RNA (>$10^6$ cDNA copies/ml) were detected in nasal wash samples collected from both species one day post-challenge (dpc). In rhesus macaques, viral RNA in nasal washes peaked at 2 dpc at $6.9 \pm 2.3 \times 10^7$ cDNA copies/ml and levels remained between $2.9 \times 10^5$ and $4.8 \times 10^7$ cDNA copies/ml until 11 dpc, before decreasing to ≤$1.8 \times 10^4$ cDNA copies/ml by 18 dpc. Cynomolgus macaques displayed a similar pattern of viral RNA burden in nasal wash samples with peak levels detected at 3 dpc. However, levels later in infection remained higher with titres of $2.0 \times 10^6$ cDNA copies/ml at 15 dpc and $1.6 \times 10^5$ cDNA copies/ml at 19

dpc (Fig. 3a). These results were in line with the viral subgenomic RNA (sgRNA) that was greatest for rhesus macaques at 2 dpc and cynomolgus macaques at 3 dpc (Fig. 3b).

Viral load in throat swabs largely mirrored that in nasal washes with peak levels early in infection, although, overall titres were lower. This was most notable at 2 dpc (Fig. 3c). In rhesus macaques, viral RNA was detected above the lower limit of quantification (LLOQ—$2.66 \times 10^3$ copies/ml) in all but one animal between 1 and 3 dpc and remained ≥$1.5 \times 10^4$ copies/ml for all animals between 4 and 9 dpc before falling and remaining below the assay's lower limit of detection (LLOD) from 11 to 18 dpc. Throat swabs from cynomolgus macaques contained higher levels of viral RNA early in infection (1–3 dpc) and remained ≥$4.5 \times 10^4$ copies/ml for all animals between 4 and 9 dpc. The viral subgenomic RNA (sgRNA) was highest for rhesus macaques at 1 and 3 dpc and for cynomolgus macaques at 1, 5 and 6 dpc (Fig. 3d), which corresponded to the highest level of total viral RNA detected.

Viral load in BAL samples echoed URT samples, with high (≥$9.8 \times 10^6$ copies/ml) levels in both species at 4 and 5 dpc, dropping to ≤$2.4 \times 10^4$ copies/ml and ≤$1.9 \times 10^4$ copies/ml at 14 dpc and 15 dpc in rhesus and cynomolgus macaques, respectively (Fig. 3e). These results were mirrored by the viral subgenomic RNA with $1.7 \times 10^5$ copies/ml (sgRNA) at 5 dpc for rhesus and $4.8 \times 10^4$ copies/ml (sgRNA) at 4 dpc for cynomolgus macaques (Fig. 3f).

Virus shedding from the gastrointestinal tract was assessed by RT-qPCR performed on rectal swab samples. In rhesus macaques, low levels of viral RNA were detected from 1 to 9 dpc. In cynomolgus macaques, viral RNA was similarly detected at a low level in rectal swabs from 1 to 9 dpc. However viral RNA levels above the LLOQ were detected at both 3 and 5 dpc in cynomolgus macaques in comparison to 2 and 3 dpc in rhesus macaques (Fig. 3g).

Viral RNA was detected at only two timepoints after challenge in whole blood samples, although below the LLOQ, and remained below the LLOQ throughout the study (Fig. 3h). In rhesus macaques, viral RNA was detected in one animal at 3 dpc, whilst in cynomolgus macaques, viral RNA was detected in two animals at 6 dpc.

Viral RNA was detected in the nasal cavity, lung and trachea early in infection at 4 and 5 dpc. Subgenomic RNA was also detected in the nasal cavity of cynomolgus macaques at four and 5 dpc. In contrast, subgenomic RNA was detected above LLOQ for rhesus at 4 and 5 dpc, and only at 5 dpc for cynomolgus macaques. Tonsil samples were all below the LLOQ for all timepoints. (Supplementary Fig. 1).

Samples collected from the upper respiratory tract at each study time point were evaluated using a Vero cell-based plaque assay for the presence of viable virus. Viable virus was successfully recovered from two of the six rhesus macaques and three of the six cynomolgus macaques on the first occasion of sampling (1–3 dpc). In all cases, recovery was below 100 pfu/mL based on a figure of under 10 plaques in a single well. Virus was not recovered from any nasal wash samples collected from 4 dpc onwards.

**Pathological changes.** Gross pathological changes were found in the lungs of all animals from both species and sexes euthanised 4/5 days after challenge and comprised multiple areas of mild to moderate consolidation distributed in cranial and caudal lobes (Supplementary Fig. 2). At 14/15 and 18/19 dpc, only small areas of consolidation were observed. Macroscopic changes, considered to be associated with infection, were not observed in any other organ analysed in this study at any time point.

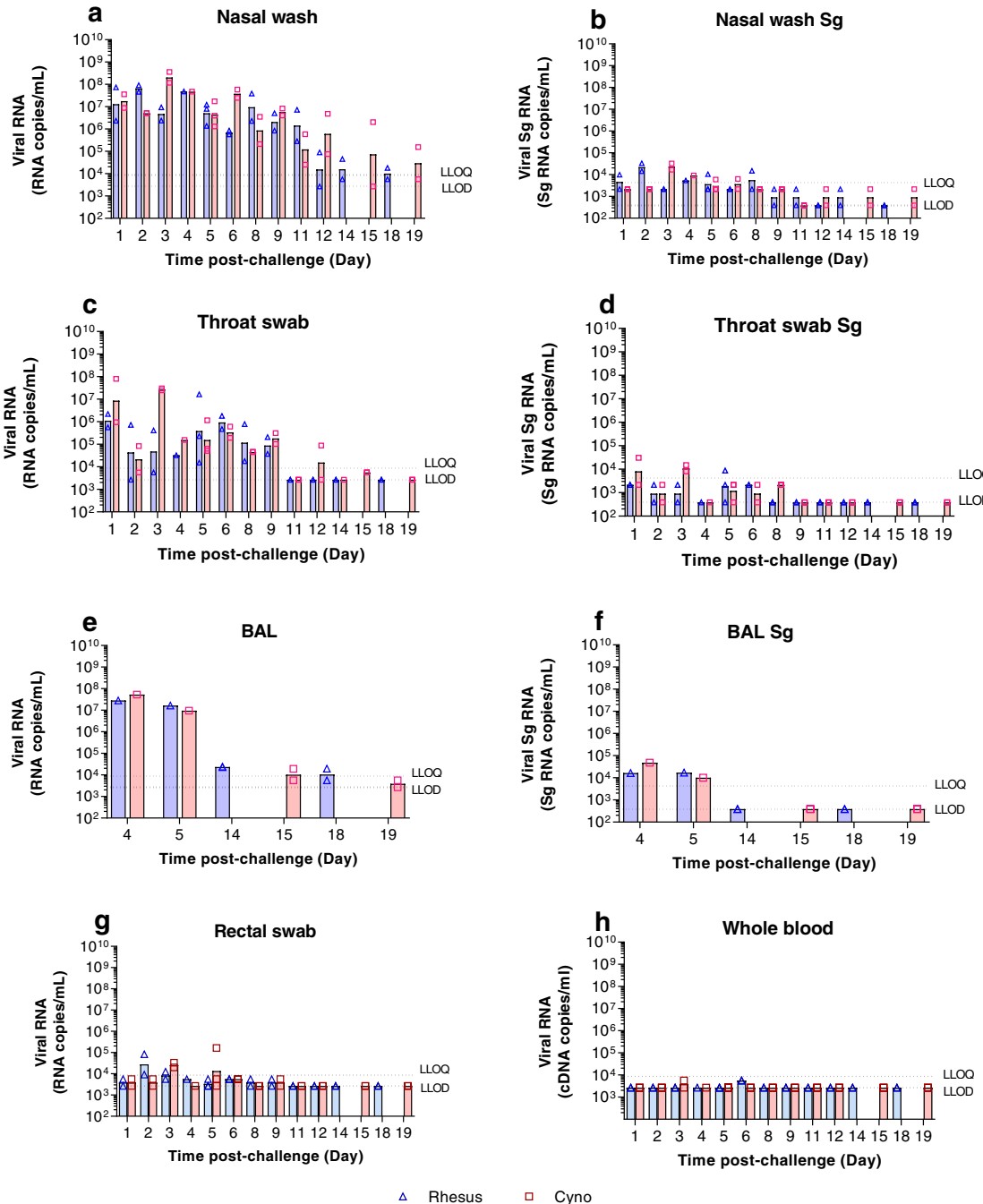

**Fig. 3 Viral RNA detected by RT-qPCR.** Viral load presented as the geometric mean of RNA copies/mL or individual values, with individual data points overlaid in rhesus macaques (blue) and cynomolgus macaques (red) in (**a**) nasal wash total RNA, (**b**) nasal wash Sg RNA, (**c**) throat swab total RNA, (**d**) throat swab Sg RNA (**e**) bronchoalveolar lavage (BAL) total RNA, (**f**) bronchoalveolar lavage (BAL) Sg RNA (numbers indicate days post-challenge the NHP was euthanised), (**g**) rectal swab total RNA, (**h**) whole blood total RNA. $n = 2$ macaques per group/species at 1, 2, 3, 6, 8, 9, 11, 12, 14, 15, 18 and 19 dpc; $n = 1$ at 4 dpc; $n = 3$ at 5 dpc (for BAL; $n = 1$ at 4 and 5 dpc and $n = 2$ at 14,15, 18 and 19 dpc). Bars show median values. Dashed lines highlight the LLOQ (lower limit of quantification, $8.57 \times 10^3$ copies/mL for total RNA and $1.29 \times 10^4$ copies/mL subgenomic RNA) and LLOD (lower limit of detection, $2.66 \times 10^3$ copies/mL for total RNA, $1.16 \times 10^3$ copies/mL for subgenomic RNA). Positive samples detected below the LLOQ were assigned the value of $5.57 \times 10^3$ copies/mL for total RNA and $12.86 \times 10^4$ copies/mL subgenomic RNA. Viral RNA was not detected in naïve animals. Samples for RT-qPCR and sgPCR were assayed in duplicate against a standard curve in triplicate.

Histological changes in the lungs of all twelve animals from both species, consistent with infection with SARS-CoV-2, were observed. The changes were most prominent at 4/5 dpc and thereafter were less severe, indicating resolution of the more acute changes observed at early time points.

Four and five days after challenge, the lung parenchyma in the cynomolgus macaques was comprised of multifocal to coalescing areas of pneumonia, surrounded by unaffected parenchyma. Overall, alveolar necrosis was a prominent feature in the affected areas, characterised by individual, shrunken, eosinophilic cells in

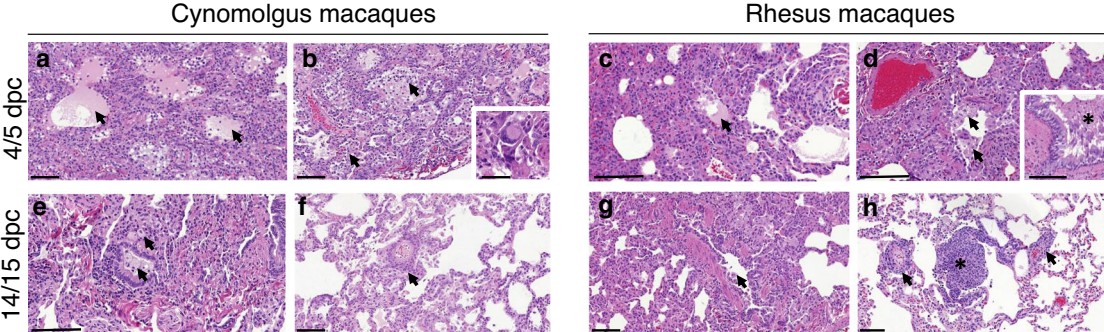

Cynomolgus macaques Rhesus macaques

**Fig. 4 Histopathological changes in cynomolgus and rhesus macaques during SARS-CoV-2 infection.** Areas of alveolar necrosis observed in cynomolgus macaques at 4/5 dpc with shrunken, eosinophilic cells within the alveolar walls (**a**, **b**), together with alveolar oedema (**a**, arrows; bar = 100 µm), type II pneumocyte hyperplasia and expanded alveolar spaces with inflammatory cell infiltration (**b**, arrows; bar = 100 µm). Occasional multinucleated cells are observed (**b**, insert; bar = 20 µm). Similar histopathological changes observed in rhesus macaques, including alveolar necrosis and areas with patchy alveolar oedema (**c**, arrow; bar = 100 µm), and accumulatios of alveolar macrophages (**d**, arrow; bar = 100 µm) and bronchial exudates (**d**, insert; bar = 50 µm). Histopathological changes with less severity observed at 14/15 dpc in cynomolgus macaques, with infiltration of mononuclear cells within alveolar spaces and bronchiolar lumen (**e**, arrows; bar = 100 µm) and perivascular cuffing (**f**, arrow; bar = 100 µm). Bronchiole regeneration (**g**, arrow; bar = 100 µm) and perivascular/peribronchiolar cuffing observed in rhesus macaques at 14/15 dpc (**h**, arrows; bar = 100 µm), together with BALT proliferation (**h**, *; bar = 100 µm). Representative images from 6 slides per animal at each time point.

alveolar walls, with pyknotic or karyorrhectic nuclei (Fig. 4a). In these areas, alveolar spaces were often obliterated by collapse of the thickened and damaged alveolar walls which contained mixed inflammatory cells (Fig. 4a,b); or had obvious, alveolar type II pneumocyte hyperplasia (alveolar epithelialisation), as well as expanded alveolar spaces (Fig. 4b). Alveolar spaces were expanded and filled with fibrillar to homogenous, eosinophilic, proteinaceous fluid (alveolar oedema) (Fig. 3a), admixed with fibrin, neutrophils, enlarged alveolar macrophages, few lymphocytes and detached type II pneumocytes. In distal bronchioles, degeneration and sloughing of epithelial cells were present, with areas of attenuation; in alveolar walls associated with bronchiolo-alveolar junctions, foci of plump, type II pneumocytes were noted, representing regeneration.

In the larger airways occasional, focal, epithelial degeneration and sloughing were observed in the bronchial epithelium, with evidence of regeneration, characterised by small, basophilic epithelial cells. Low numbers of mixed inflammatory cells, comprising neutrophils, lymphoid cells, and occasional eosinophils, infiltrated bronchial and bronchiolar walls. In the lumen of some airways, mucus admixed with degenerative cells, mainly neutrophils and epithelial cells, was seen. Occasionally, multinucleated cells, characterised as large, irregularly shaped cells with prominent, eosinophilic cytoplasm and multiple round nuclei were observed (Fig. 4b, insert).

Pathological changes consistent with those described for cynomolgus macaques were present in the lungs of rhesus macaques. In the parenchyma, multifocal expansion and infiltration of alveolar walls by inflammatory cells were noted (Fig. 4c). Furthermore, in these areas, alveolar necrosis was observed with patchy alveolar oedema and accumulations of alveolar macrophages (Fig. 4c). In the bronchi and bronchioles, similar changes to those described for cynomolgus macaques were seen (Fig. 4d).

Changes were less severe in all four animals examined at 14/15 dpc. In the cynomolgus macaques, patchy infiltration of mainly mononuclear cells in the alveolar walls, with occasional similar cells within alveolar spaces, and parenchymal collapse, were seen (Fig. 4e). Mononuclear cells, primarily lymphocytes were also noted surrounding and infiltrating the walls of blood vessels (Fig. 4f) and airways. Furthermore, there was an increased prominence of bronchial-associated lymphoid tissue (BALT) was noted. In the lungs of rhesus macaques, changes in the alveoli and

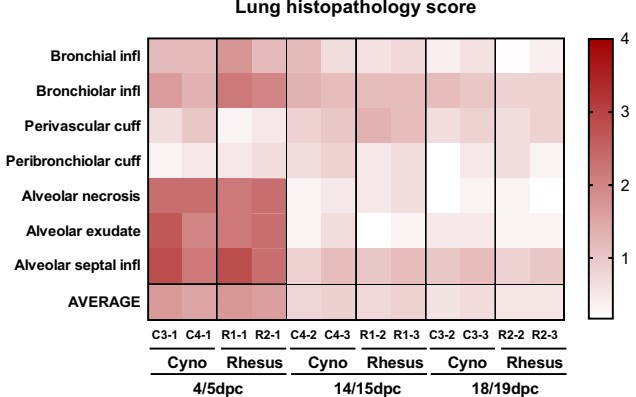

**Lung histopathology score**

**Fig. 5 Lung histopathology scores.** Heatmap showing the scores for each lung pathology parameter and the average score for each animal culled at 4/5, 14/15 and 18/19 dpc. Severity ranges from 0 to 4: 0 = none; 1 = minimal; 2 = mild; 3 = moderate and 4 = marked/severe.

BALT were similar in appearance and frequency to those described in the cynomolgus macaques, and perivascular lymphocytic cuffing of small vessels, characterised by concentric infiltrates of mononuclear cells, was also seen occasionally (Figs. 4g and 4h). By day 18/19, the changes were similar but less frequent to those described at day 14/15 in all four animals.

Using the subjective histopathology scoring system, scores were higher in both macaque species at 4/5 dpc compared to 14/15 and 18/19 dpc, mostly due to higher scores in the alveolar damage parameters observed at the early time point (Fig. 5).

The presence of viral RNA was observed in the lungs from all animals at 4/5 dpc by in situ hybridisation (ISH). Prominent staining of small foci of cells containing SARS-CoV-2 viral RNA, was observed within pneumocytes and interalveolar septa, concomitant with microscopic changes in cynomolgus macaques (Fig. 6a). Staining was not seen in cells or fluid within the alveolar spaces. Positive mononuclear cells were also observed rarely in the bronchus-associated lymphoid tissue (BALT) (Fig. 6a, insert). Small foci of cells staining positive for viral RNA were observed at a low frequency in the rhesus macaques within the alveolar lining

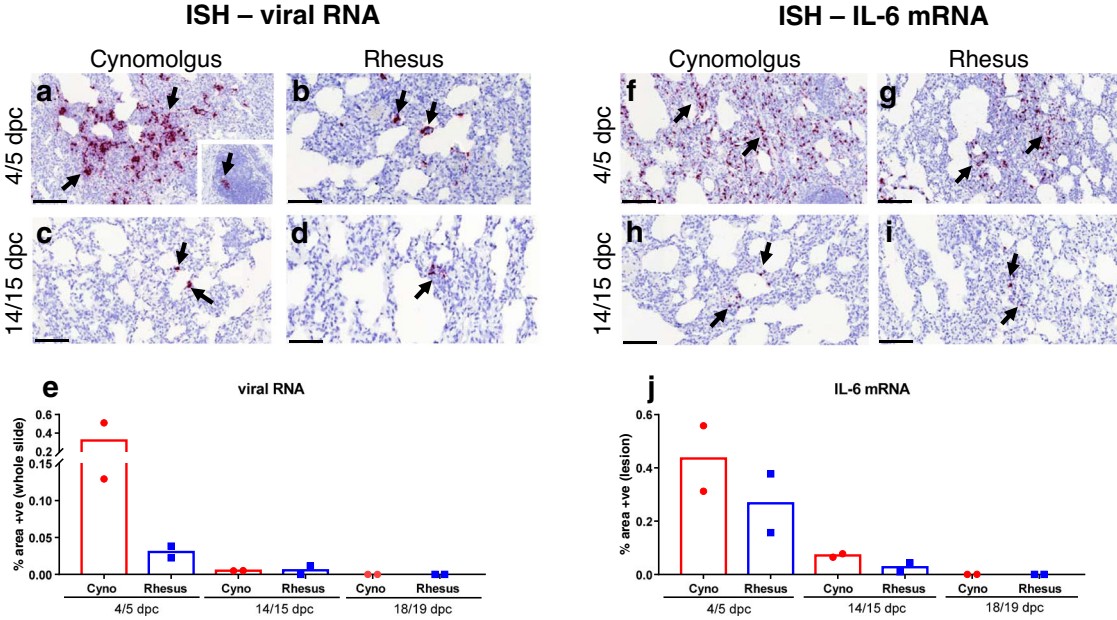

**Fig. 6 Presence of viral RNA and IL-6 by ISH in cynomolgus and rhesus macaques during SARS-CoV-2 infection.** ISH detection of abundant viral RNA (RNAScope, red chromogen) within the areas of pneumonia (**a**, arrows) and occasionally in the BALT (**a**, insert, arrow) in cynomolgus (**a**, **b**) and rhesus macaques (**b**, arrows) at 4/5 dpc. Small amount of viral RNA in the interalveolar septa from cynomolgus (**c**, arrows) and rhesus macaques (**d**, arrows) at 14/15 dpc. Image analysis of positively stained area in RNAScope labelled sections for viral RNA (**e**, whole slide); $n = 2$ macaques per species and time point; bars represent median values. Abundant presence of IL-6 mRNA in the areas of pneumonia from cynomolgus (**f**, arrows) and rhesus macaques (**g**, arrows) at 4/5 dpc. Small amount of IL-6 mRNA positive cells within the interalveolar septa from cynomolgus (**h**, arrows) and rhesus macaques (**i**, arrows) at 14/15 dpc. Image analysis of positively stained area in RNAScope labelled sections for IL-6 mRNA (**j**, areas of lesion); $n = 2$ macaques per species and time point; bars represent median values. Bars in micrographs=200 μm.

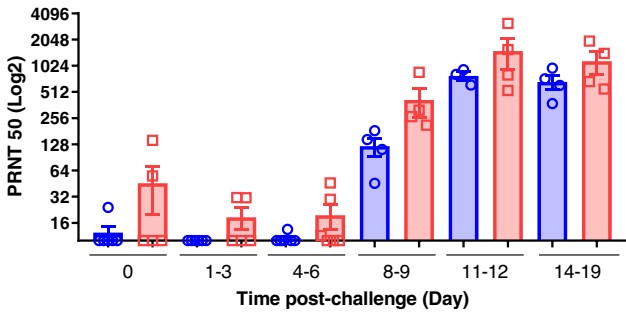

**Fig. 7 Neutralising antibodies in serum measured by Plaque reduction neutralisation test (PRNT₅₀).** Serum neutralisation titres as reciprocal highest dilution resulting in an infection reduction of >50% in samples (PRNT₅₀) pre-challenge and at 1–3, 4–6, 8–9, 11–12 and 14–19 days post-challenge in rhesus macaques (blue) and cynomolgus macaques (red). For rhesus macaques: $n = 6$ at 0, 1–3 and 4–6 dpc; $n = 4$ at 8–9 and 14–19 dpc; $n = 3$ at 11–12 dpc. For cynomolgus macaques: $n = 6$ at 4–6 dpc; $n = 5$ at 0 and 1–3 dpc; $n = 4$ at 8–9; 11–12 and 14–19 dpc. Bars indicating group mean ±standard error with PRNT₅₀ determined for individual animals shown as circles and squares respectively. Neutralising antibodies were observed at 8–9 dpc at low levels, increasing from 11 dpc onwards, with higher values in cynomolgus macaques compared to rhesus.

and interalveolar septa of both animals at 4/5 dpc, concomitant with microscopic changes (Fig. 6b).

Viral RNA was detected in only a few individual cells in both groups of animals at 14/15 dpc (Figs. 6c, d). By day 18/19, viral RNA was not detected by ISH. Overall, there was a high presence of viral RNA at 4/5 dpc which was more pronounced in the cynomolgus macaques; by contrast, only very few positive cells were observed at 14/15 dpc and none at 18/19 dpc. (Fig. 6e).

Viral RNA was observed in scattered epithelial cells in areas of the upper respiratory tract (nasal cavity, larynx and trachea) of all animals at 4/5 dpc, and not associated with lesions.

Abundant numbers of cells expressing IL-6 mRNA were observed within the pulmonary lesions, with only few positive scattered cells in the healthy parenchyma.

IL-6 mRNA was abundant within the lesions in cynomolgus (Fig. 6f) and rhesus macaques (Fig. 6g) at 4/5 dpc, which was slightly more pronounced in cynomolgus macaques (Fig. 6j). The expression of IL-6 mRNA was less pronounced in both cynomolgus (Fig. 6h) and rhesus macaques (Fig. 6i) at 14/15 dpc, with no significant expression observed at 18/19 dpc.

In the liver, microvesicular, centrilobular vacuolation, consistent with glycogen, together with, small, random, foci of lymphoplasmacytic cell infiltration were noted rarely. This is considered to represent a mild, frequently observed background lesion. Remarkable changes were not observed in any other tissue. Viral RNA staining was seen only at 4/5 dpc, in occasional, absorbing epithelial and goblet cells in the small and large intestine. It was not observed in any other tissue examined.

**Antibody responses to SARS-CoV-2 infection.** Low levels of neutralising antibody were detected by plaque reduction neutralisation test (PRNT) assay in both cynomolgus and rhesus macaques for the first 8–9 days post-challenge. From day eleven or twelve, both species showed high neutralising antibody titres which continued at the later time point (14–19 dpc). The neutralisation titres at the later time points were generally higher in the cynomolgus macaques, although greater variability in titres between different animals was seen for this species (Fig. 7).

Seroconversion to viral antigens Spike trimer, Receptor Binding Domain (RBD) and Nucleoprotein were evaluated by ELISA following infection. Specific antibodies against

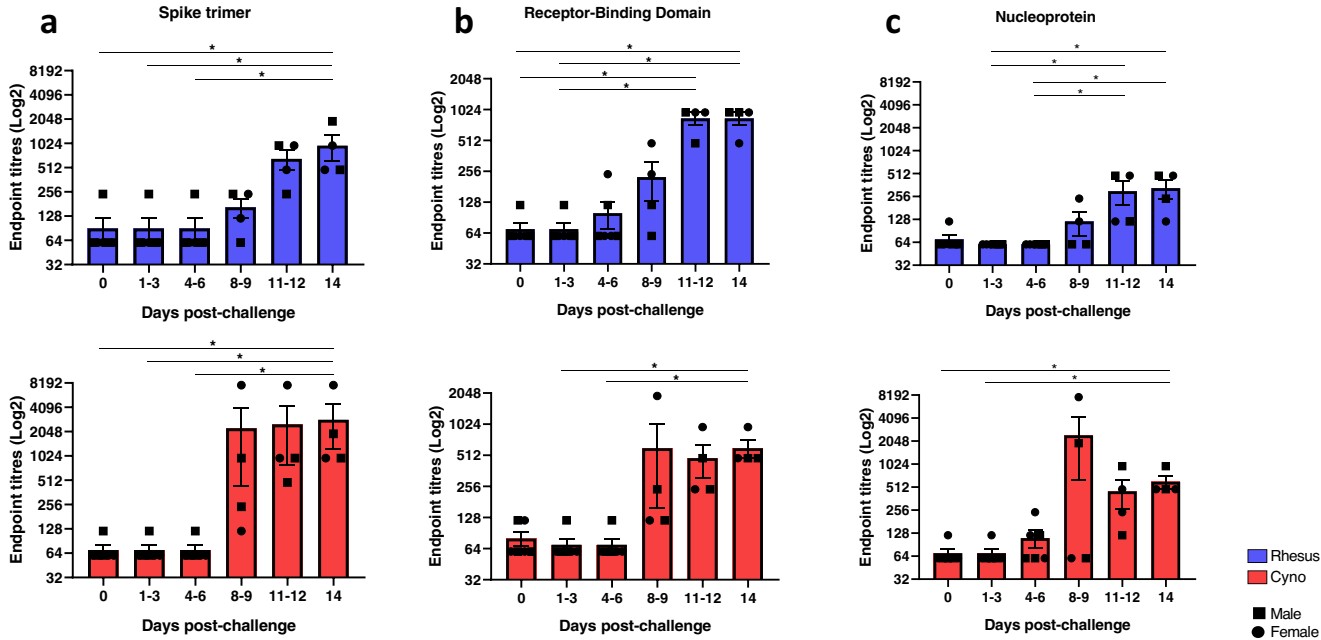

**Fig. 8 SARS-CoV-2-specific IgG antibodies measured by ELISA in naïve and SARS-CoV-2 infected macaques.** Spike- (**a**), Receptor-Binding Domain- (**b**) and Nucleoprotein- (**c**) specific IgG antibodies measured in sera of rhesus and cynomolgus macaques. Sera were collected from uninfected animals (day 0) or 1–3, 4–6, 8–9, 11–12 and 14–19 days following SARS-CoV-2 infection. $n = 6$ at 0, 1–3 and 4–6 dpc; $n = 4$ at 8–9, 11–12 and 14 dpc. Bars show the group mean±SEM with an endpoint titre determined for each individual animal shown as squares for males and dots for females. *$p \leq 0.05$ (Kruskal–Wallis one-way ANOVA, two-sided). Experiment performed in duplicates.

SARS-CoV-2 were detected in serum collected from both rhesus and cynomolgus macaques from 8 to 9 dpc onwards (Fig. 8).

**Frequency of antigen-specific IFN-γ secreting cells measured by ELISpot.** Cellular immune responses induced by SARS-CoV-2 challenge were measured in peripheral blood mononuclear cells (PBMCs) using an ex vivo IFN-γ ELISpot assay and compared to responses measured in uninfected (naïve) age and species matched control animals (Fig. 9). Interrogation of IFN-γ spot forming units (SFU) measured in response to stimulation with overlapping 15-mer spike protein-peptide pools indicated that peptides spanning the breadth of the SARS-CoV-2 spike protein sequence induced cellular immune responses in infected cynomolgus macaques; whereas, peptide pool (PP) sequences 2-4 and 8 were most immunogenic in rhesus macaques (Fig. 9a, b).

In rhesus macaques, the IFN-γ SFU measured following stimulation with spike protein peptide megapools (MP) 1–3 did not differ significantly between animals euthanised at either the day 4–5 (early) or the day 14–19 (late) post-infection time-point in comparison to SFU frequencies measured in the naïve control animals. However, comparison of the summed MP 1–3-specific response indicated that significantly higher SFU frequencies were present in the animals euthanised at the later time-point ($P = 0.01$) (Fig. 9c). By contrast, spike protein MP1-, MP2, and summed MP-specific responses measured in cynomolgus macaques were all significantly higher than in naïve control animals at the later post-infection time-point ($p = 0.03$, $p = 0.01$, $p = 0.01$) (Fig. 9d).

In general, there was a trend for spike protein peptide-specific IFN-γ SFU frequencies measured in PBMC samples collected from cynomolgus macaques to be greater than those detected in rhesus macaques, although these differences did not reach statistical significance.

Spike peptide-specific IFN-γ SFU frequencies measured in mononuclear cells isolated from lung and spleen samples revealed

a trend for local cellular immune responses to be greatest in the animals euthanised at the day 14 to19 post-infection time point, but also that there was substantial variability within the groups at this stage of infection (Fig. 9e).

*Composition and functional profile of the cell-mediated immune response.* To explore changes in the composition of the cellular immune compartment following SARS-CoV-2 infection, immunophenotyping flow cytometry assays were applied to PBMCs and lung mononuclear cells (MNC) samples collected at necropsy and from PBMCs collected from age and species matched uninfected (naïve) animals. Comparison of CD4+ and CD8+ T-cell frequencies indicated that the proportion of CD8+ T-cells was greater in cynomolgus macaques prior to, and also after infection when the frequency of the CD4+ subset increased but remained secondary to the CD8+ population in peripheral blood and lung MNC (Fig. 9f). Similarly, the frequency of γδ T-cells in peripheral blood was higher in cynomolgus macaques but remained consistent following infection, whereas γδ T-cell frequencies appeared more variable in the lung MNC samples collected from animals euthanised at the early and late post-infection time point (Fig. 9g, h). Monocyte subtypes were characterised as classical, non-classical or intermediate by expression of CD14+ and CD16+. This revealed an increased frequency of 'transitional' CD14+ CD16+ and 'non-classical' CD14− CD16+ monocyte subsets in PBMCs collected at the later post-infection time point in both rhesus and cynomolgus species. In contrast, the immunomodulatory non-classical monocyte population was more abundant in lung tissue samples collected from rhesus macaques euthanised early after infection in comparison to the later time points or to cynomolgus macaques (Fig. 9i). Cytotoxic and immunomodulatory natural killer (NK) cell populations were identified within the CD3−, CD159a+ lymphocyte population based on the expression on CD16 and CD56, respectively. Immunomodulatory (CD56+) NK cell populations were detected at higher frequency in the lung and PBMC of infected macaques

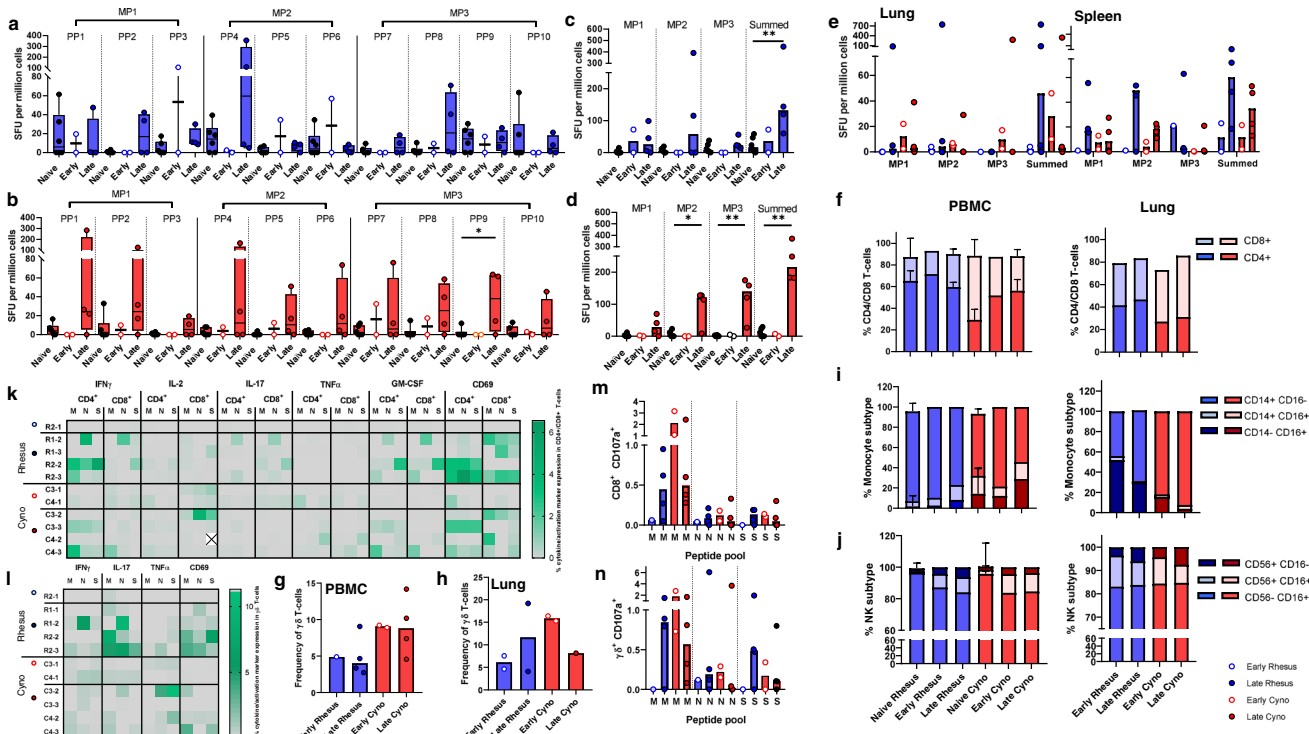

**Fig. 9 Cellular immune responses to SARS-CoV-2. a, b** IFNγ SFU measured in PBMCs and stimulated with spike protein peptide pools (PP) peptide in (**a**) rhesus and (**b**) cynomolgus macaques. PBMC samples were isolated from uninfected animals (naïve) or at early (days 4 and 5) and late (days 14-19) time-points following SARS-CoV-2 infection. Box plots show the group median +/− inter-quartile range, with minimum and maximum values connected by whiskers. Two-tailed Mann–Whitney U-test carried out to compare pre and post-SARS-Cov2 infection where *$p ≤ 0.05$, **$p ≤ 0.01$. **c, d** IFNγ SFU measured in PBMC in response to spike protein megapools (MP) in (**c**) rhesus and (**d**) cynomolgus macaques or, (**e**) in mononuclear cells isolated from lung and spleen. Bars show the group median with SFU measured in individual animals shown as dots. Rhesus macaques summed MP naïve vs late time point $p = 0.01$. Cynomolgus macaques naïve vs PP9 $p = 0.03$, naïve vs MP2 $p = 0.03$, naïve vs MP3 $p = 0.01$, naïve vs summed $p = 0.01$. Biologically independent animal samples for (**a–d**); PBMCs: naïve $n = 6$, early $n = 2$, late $n = 4$ Biologically independent animal samples for (**e**); lung and spleen: early $n = 2$, late $n = 4$, (**f–j**) Frequency of major lymphocyte and monocyte cell populations quantified by immunophenotyping assay (**f–h**) CD4+, CD8+ and γδ T-cell frequencies in PBMCs and lung cells, (**i**) Monocyte subtype frequency in PBMCs and lung MNCs, (**j**) Natural killer (NK) cell subset frequency in PBMCs and lung MNCs. Stacked bars show the group median with 95% confidence intervals. Biologically independent animal samples for (**f–j**); PBMC: Naïve rhesus $n = 8$, early rhesus $n = 1$, late rhesus $n = 2$, naïve cyno $n = 7$, early cyno $n = 2$, late cyno $n = 2$. Lung: early rhesus $n = 2$, late rhesus $n = 3$, early cyno $n = 2$, late cyno $n = 2$. **k–n** Intracellular cytokine staining data. **k–l** Cytokine and activation marker detection in CD4+, CD8+ and γδ T-cells in PBMCs stimulated with M, N and S peptide pools. **m–n** CD107a expression in CD8+ and γδ T-cells in PBMCs. Bars show the group median with cell frequencies measured in individual animals shown as dots.

in comparison to naïve control animals, indicating a potential proinflammatory role for this innate lymphoid cell subset in SARS-CoV-2 infection (Fig. 9j).

To explore the functional profile of T-cell populations, PBMCs were stimulated with peptide pools spanning the SARS-CoV-2 membrane (M), nucleocapsid (N) or spike (S) proteins, and the production of the cytokines IFN-γ, IL-2, TNF-α, IL-17 and GM-CSF along with the activation marker CD69 and degranulation marker CD107a measured by intracellular cytokine staining. Cytokine-producing CD4 and CD8 T-cells were detected in both rhesus and cynomolgus macaques in response to stimulation with M, N and S peptide pools. Proinflammatory (IFN-γ or GM-CSF producing) T-cells were primarily detected at the later post-challenge time point, although low frequencies of IL-2 producing CD8+ cells were detected in PBMC samples collected from cynomolgus macaques in the early post-challenge samples (Fig. 9k). The frequency of Th17 and TNF-α expressing cells differed between the species with IL-17 producing CD4 and CD8 T-cells more prevalent in rhesus macaques, whereas TNFα expression was detected more frequently in cynomolgus macaques. Similarly, cytokine production measured in the γδ T-cell

population indicated a trend for greater IL-17 production in PBMCs isolated from rhesus macaques, although low frequencies of IFN-γ and IL-17 producing γδ T-cells were also detected in cynomolgus macaques euthanised at the early post-infection time point indicating that unconventional T-cell populations play a role in the early immune response to SARS-CoV-2 infection (Fig. 9l). Peptide-specific expression of the degranulation marker CD107a was assessed as a measure of cell mediated cytotoxicity. CD107a was detected on CD8 and γδ T-cells in both rhesus and cynomolgus macaques, was most potently induced by stimulation with the M protein peptide pool and detected at higher frequency in cynomolgus macaques at the early post-infection time points (Fig. 9m, n). In addition to the above functional parameters, antigen-specific expression of the activation marker CD69 was assessed to provide a measure of the overall activation status and SARS-CoV-2 antigen-reactogenicity of T-cell subsets following infection. In general, CD69 expression on CD4, CD8 and γδ T-cell populations was higher at the later post-infection timepoints (Fig. 9k-l) and expression levels agreed with the detection of cytokine production in the corresponding T-cell subsets. However, instances of CD69 expression were also apparent in M, N

and S peptide stimulated samples, in which there was an absence of cytokine or degranulation marker detection, indicating that these activated cells may have exerted antigen-specific functions outside of the parameters measured by our ICS assay.

**Gene expression and gene network comparison of the host response to SARS-CoV-2.** Gene expression analysis by defining mRNA profiles in clinical samples can be used to characterise the host response to infection. For example, this approach has been used to analyse blood samples to differentiate patients with acute Ebola virus disease who went on to survive or die[16] or to characterise Zika virus infection in the placenta during different trimesters[17]. Therefore, we used this approach to compare SARS-CoV-2 infection between the cynomolgus and rhesus macaques. In this case, RNAseq was used to identify and quantify mRNA in blood samples taken longitudinally during infection. Overall, the number of gene structures in the Ensembl genomes that used as references for alignment, were 32,386 for the rhesus macaques and 29,324 in cynomolgus macaques. 24,300 genes defined as one to one orthologous in each species were used in a gene-network analysis to compare the host response at a global level for each species (Supplementary Fig. 3). Genes with a similar function were clustered together into consensus modules to allow different traits (sex of the animal and days post-infection) to be compared. The data was also separated into consensus and opposite signs between the cynomolgus and rhesus macaques. In general, the consensus module showed similarity in the response to SARS-CoV-2 between cynomolgus and rhesus macaques. For example, the inflammatory response, biotic stimulus and interferon were similar for both species in that the response is negatively related to the dpc. I.e., the response occurred at the start of infection and then wanes with length of infection. Similarly, this pattern was seen with interferon production and response to virus for both species. One exception was in the positive regulation of cytokine production where the response in cynomolgus macaques had a stronger negative response than rhesus macaques. But for both species' cytokine production was active at the beginning of infection and becomes weaker at later time points. There was some evidence of potential sex differences, although the *n* number maybe too low to draw specific conclusions (Supplementary Fig. 3).

## Discussion
We have shown in a head to head comparison that the consequences of challenge with SARS-CoV-2 in rhesus and cynomolgus macaques are similar and in line with outcomes described in studies conducted individually in either, rhesus[11,12,18,19] or cynomolgus macaques[13,20]. Uniquely, our conclusion that these two species respond similarly to SARS-CoV-2 infection, is supported by our transcriptional analysis which provides a global view on host gene expression.

The clinical manifestations in human COVID-19 patients range from asymptomatic to severe[21–23]. In our study, we have observed that SARS-CoV-2 induced features characteristic of COVID-19 identified on CT scans in the absence of clinical signs in both rhesus and cynomolgus macaques. The lack of clinical signs observed agrees with other reports that used the IN and IT routes of challenge delivery[11,13,18,20]. The slightly increased levels of clinical changes previously described in rhesus[12,24,25] may be due to the use of the ocular and oral routes in addition to IN and IT for challenge and the potential impact of additional in-life sampling. CT provided the only clinical measure that identified abnormalities in vivo consistent with COVID-19 and is therefore a critical tool for evaluation of disease burden following experimental infection. Features characteristic of COVID-19 in human

patients, such as ground glass opacity and consolidation were identified on CT scans collected from both SARS-CoV-2 challenged rhesus and cynomolgus macaques in line with reports from studies describing SARS-CoV-2 infection in either, rhesus[24], or cynomolgus macaques[20] supporting a role for both species as models of human SARS-CoV-2 induced disease.

The pattern of viral shedding from the URT (peak day one to three with subsequent decline to undetectable), intermittent low recovery from the gastrointestinal tract and the absence of detection above the LLOQ in the blood described across both species is similar to what is observed in humans with asymptomatic/mild COVID-19 (shedding without clinical signs and/or after the resolution of clinical course) and reflect previous reports of SARS-CoV-2 infection in rhesus[11,12,18,19] and cynomolgus macaques[13,20]. Similar to these studies, virus shed from the URT was detected by RT-qPCR very shortly after challenge, which could suggest the presence of residual challenge material rather than shedding of the newly replicated virus resulting from infection. However, subgenomic RNA was detected at these early timepoints suggesting viral replication had indeed occurred in both species. Similar to Rockx et al.[13] and Munster et al.[12] we detected live virus, albeit at low levels in samples from the URT. Positive samples were restricted to early timepoints post-infection-challenge, which suggests some level of new virus replication which is also supported by the level of virus detected using RNA probes conducted on samples collected from animals euthanised 4/5 dpc[13].

Histopathological changes were comparable in both animal species, and closely resembled that seen in human cases following a mild/moderate clinical course[26–28]. Typical changes of acute respiratory viral infection were observed, with alveolar necrosis and type II pneumocyte hyperplasia seen on microscopy together with interstitial lymphoid infiltrates, focally showing perivascular cuffing. These histopathological lesions are compatible to those observed in human patients, although some features seen in humans, such as thromboembolic changes, were not present in the macaques[29]. The pathology at 14/15 and 18/19 dpc showed signs of resolution. We have described here a histopathology scoring system that allows us to actually quantify the severity of lesions in the airways and parenchyma of lung tissue sections from SARS-CoV-2 infected macaques. We have used this system to compare quantitatively the lung histopathology observed in pre-clinical vaccine studies. It will also be applicable to assessing the efficacy of therapeutics in these NHP COVID-19 models. We have also described herein an upregulation of local interleukin-6 transcription at the mRNA level within the pulmonary lesions at the early time points of infection in both species. The induction of a proinflammatory cytokine storm has been described in human COVID-19 patients, with IL-6 levels significantly elevated and associated with the disease severity[30,31].

The development of neutralising titres of specific antibodies is important for the control of infection and viral transmission and is a commonly reported feature in COVID-19 patients, although neutralising antibody alone is not considered sufficient for protection against severe disease[32]. IgG seroconversion occurred in both macaque species from days 8-9 post-infection and therefore follows a similar kinetic to serology profiles measured in COVID-19 patients[33], indicating that both species offer representative models for the investigation of SARS-CoV-2 related humoral immunity.

In keeping with the generally mild pathology and limited evidence of viral replication or persistence reported in both macaque species, we detected little evidence in cellular immune profiles of the immune dysregulation associated with severe COVID-19 disease in humans[34]. Immunological features were more typical of those reported in milder infections and convalescent patients

and included changes in the frequency of CD4 and CD8 T-cell populations[35] as well as increased frequencies of immunomodulatory NK[36] and monocyte subsets[37].

The role of T-cells in SARS-CoV-2 immunity is not fully defined, although, it is clear that CD4 and CD8 memory T-cells are present in COVID-19 convalescent patients and those previously exposed to related coronaviruses[38]. CD4 T-cell dependent mechanisms of protection have been demonstrated in small animal models of SARS-CoV infection[39], T-cells are likely to play an important role in the development of neutralising antibodies[40] and the clearance of infection through cell-mediated cytotoxicity[32]. Our findings confirm that proinflammatory and cytotoxic T-cells are induced by SARS-CoV-2 infection in both rhesus and cynomolgus macaques to a similar extent, and that there is T-cell reactogenicity in both species to all three of the SARS-CoV-2 antigens included in our assays. This included the detection of antigen-specific immune responses directed toward peptide-epitopes spanning the S protein sequence, an antigen incorporated into several of the novel vaccine candidates currently under investigation[41], thus demonstrating the value of the macaque model for immunogenicity testing of novel SARS-CoV-2 vaccine candidates. Additionally, in a comparative study of colonisation of NHPs by group A *Steptococcus pyogenes* it was reported that cynomolgus developed a stronger antibody response compared to rhesus macaques[42], a trend that we also observed.

The sample size used in this study is relatively limited (6 rhesus and 6 cynomolgus macaques), and therefore, there are limitations to provide statistical significance for some endpoints. However, our results clearly demonstrate that both species provide infection models of SARS-CoV-2, which reflect upper and lower respiratory tract viral replication resulting in a lung injury, repair and resolution picture typical for milder forms of COVID-19 disease in humans. Both macaque models potentially represent the majority of the human population and enable evaluation of the safety and efficacy of novel and repurposed interventions against SARS-CoV-2 using endpoints of upper and lower respiratory tract virus replication, in addition to CT and histopathology in the assessment of significant but transient lung injury. However, further work is needed to develop models that are representative of the more severe outcomes that would particularly enable the evaluation of therapies targeting host-mediated pathology associated with high levels of prolonged, pulmonary disease. Given the limitations in reproducing the range of underlying health conditions in humans that link to poorer infection outcomes (e.g. diabetes, obesity and age) in macaques, other strategies of disease enhancement such as challenge, route, dose, strain, or manipulation of the host immunity, will be required.

The limited supply of rhesus macaques is now impacting on future COVID-19 studies to support the development of vaccines and therapeutic products[43]. The potential offered by cynomolgus macaques as an appropriate model will greatly increase the international community's ability to perform these critical studies in support of pre-clinical evaluation and product licensure. Moreover, cynomolgus macaques with a Mauritian genotype have more restricted genetic variability and more limited and better-defined MHC, providing an advantage in the battle to elucidate correlates of protective immunity. These features have been of particular value in HIV vaccine research where models are established in both rhesus macaques (Indian genotype) and cynomolgus macaques (Mauritian genotype)[44]. The MHC homogeneity associated with the Mauritian cynomolgus macaque has reduced the variability between animals after vaccination and has enhanced comparisons of vaccine regimens. In addition, the improved consistency in outcome facilitates the use of fewer animals to obtain statistically significant results than would be required if more genetically diverse species were to be used.

## Methods

**Animals.** Six cynomolgus macaques of Mauritian origin (*Macaca fascicularis*) and six rhesus macaques of Indian origin (*Macaca mulatta*) were used in this study (Fig. 1). Study groups comprised three males and three females of each species and all were adults aged 2–4 years and weighing between 2.89 and 4.85 kg at time of challenge. Before the start of the experiment, socially compatible animals were randomly assigned to challenge groups, to minimise bias.

Animals were housed in compatible social groups, in cages in accordance with the UK Home Office Code of Practice for the Housing and Care of Animals Bred, Supplied or Used for Scientific Procedures (2014) and National Committee for Refinement, Reduction and Replacement (NC3Rs) Guidelines on Primate Accommodation, Care and Use, August 2006[45]. Prior to challenge animals were housed at Advisory Committee on Dangerous Pathogens (ACDP) level two in cages approximately 2.5 M high by 4 M long by 2 M deep, constructed with high level observation balconies and with a floor of deep litter to allow foraging. Following challenge animals were transferred to ACDP Level three and housed in banks of cages of similar construction placed in directional airflow containment systems that allowed group housing and environmental control whilst providing a continuous, standardised inward flow of fully conditioned fresh air identical for all groups. Additional environmental enrichment was afforded by the provision of toys, swings, feeding puzzles and DVDs for visual stimulation. In addition to ad libitum access to water and standard old-world primate pellets, diet was supplemented with a selection of fresh vegetables and fruit. All experimental work was conducted under the authority of a UK Home Office approved project license (PDC57C033) that had been subject to local ethical review at PHE Porton Down by the Animal Welfare and Ethical Review Body (AWERB) and approved as required by the Home Office Animals (Scientific Procedures) Act 1986. Animals were sedated by intramuscular (IM) injection with ketamine hydrochloride (Ketaset, 100 mg/ml, Fort Dodge Animal Health Ltd, Southampton, UK; 10 mg/kg) for procedures requiring removal from their housing. None of the animals had been used previously for experimental procedures.

**Viruses and Cells.** SARS-CoV-2 Victoria/01/2020[46] was generously provided by The Doherty Institute, Melbourne, Australia at P1 after primary growth in Vero/hSLAM cells and subsequently passaged twice at PHE Porton in Vero/hSLAM cells [ECACC 04091501]. Infection of cells was with ~0.0005 MOI of virus and harvested at day 4 by dissociation of the remaining attached cells by gentle rocking with sterile 5 mm borosilicate beads followed by clarification by centrifugation at $1000 \times g$ for 10 min. Whole-genome sequencing was performed, on the P3 challenge stock, using both Nanopore and Illumina as described previously[47]. Virus titre of the challenge stocks was determined by plaque assay on Vero/E6 cells [ECACC 85020206]. Cell lines were obtained from the European Collection of Authenticated Cell Cultures (ECACC) PHE, Porton Down, UK. Cell cultures were maintained at 37 °C in Minimum essential medium (MEM) (Life Technologies, California, USA) supplemented with 10% foetal bovine serum (FBS) (Sigma, Dorset, UK) and 25 mM HEPES (Life Technologies, California, USA). In addition, Vero/hSLAM cultures were supplemented with 0.4 mg/ml of geneticin (Invitrogen) to maintain the expression plasmid. Challenge substance dilutions were conducted in phosphate buffer saline (PBS). Inoculum ($5 \times 10^6$ PFU) was delivered by intratracheal route (2 ml) and intranasal instillation (1.0 ml total, 0.5 ml per nostril).

**Clinical signs and in-life imaging by computerised tomography.** Weight and body temperature were monitored daily (Fig. 1). Nasal washes, throat and rectal swabs were taken at intervals of three days for each individual, having samples from at least two animals for each species until 5 dpc. Whole blood and serum were collected at the same time points. Nasal washes were obtained by flushing the nasal cavity with 2 ml PBS. For throat swabs, a flocked swab (MWE Medical Wire, Corsham, UK) was gently stroked across the back of the pharynx in the tonsillar area. Throat and rectal swabs were processed, and aliquots stored in viral transport media (VTM) and AVL buffer (Qiagen, Milton Keynes, UK) at −80 °C until assay.

Animals were monitored multiple times per day for behavioural and clinical changes. Behaviour was evaluated for contra-indicators including depression, withdrawal from the group, aggression, changes in feeding patterns, breathing pattern, respiration rate and cough. Prior to blood sample collection, aerosol challenge and euthanasia, animals were weighed, examined for gross abnormalities and body temperature measured (Fig. 1).

CT scans were performed at 18 dpc from the four remaining animals at this time point (Fig. 1). CT imaging was performed on sedated animals using a 16 slice Lightspeed CT scanner (General Electric Healthcare, Milwaukee, WI, USA) in the prone and supine position to assist the differentiation of pulmonary changes at the lung bases caused by gravity dependant atelectasis, from ground glass opacity caused by SARS-CoV-2. All axial scans were performed at 120 KVp, with Auto mA (ranging between 10 and 120) and were acquired using a small scan field of view. Rotation speed was 0.8 s. Images were displayed as an 11 cm field of view. To facilitate full examination of the cardiac and pulmonary vasculature, lymph nodes and extrapulmonary tissues, Niopam 300 (Bracco, Milan, Italy), a non-ionic, iodinated contrast medium, was administered intravenously (IV) at 2 ml/kg body weight and scans were collected immediately after injection and ninety seconds from the mid-point of injection. Scans were evaluated by an expert thoracic

radiologist, blinded to the animal's clinical status, for the presence of: disease features characteristic of COVID-19 in humans (ground glass opacity (GGO), consolidation, crazy paving, nodules, peri-lobular consolidation; distribution: upper, middle, lower, central 2/3, bronchocentric); pulmonary embolus and the extent of any abnormalities estimated (<25%, 25–50%, 51–75%, 76–100%).

**Post-mortem examination and histopathology.** Animals were euthanised at 3 different time-points, in groups of four (including one animal from each species and sex) at 4/5, 14/15 and 18/19 dpc (Fig. 1).

Animals were anaesthetised with ketamine (17.9 mg/kg body weight) and exsanguination was performed via cardiac puncture, followed by injection of an anaesthetic overdose (sodium pentabarbitone Dolelethal, Vetquinol UK Ltd, 140 mg/kg) to ensure euthanasia. Post-mortem examination and sample collection were performed immediately after confirmation of death.

The bronchial alveolar lavage fluid (BAL) was collected at necropsy from the right lung. The left lung was dissected prior to BAL collection and used for subsequent histopathology and virology procedures. At necropsy nasal washes, throat and rectal swabs, whole blood and serum were taken alongside tissue samples for histopathology.

Samples from the left cranial and left caudal lung lobe together with spleen, kidney, liver, mediastinal and axillary lymph nodes, small intestine (duodenum, jejunum and ileum), large intestine (caecum and colon), encephalon (cerebrum, cerebellum and brainstem), eye, trachea, larynx and nasal cavity, were fixed by immersion in 10% neutral-buffered formalin and processed routinely into paraffin wax. Nasal cavity samples were decalcified using an EDTA-based solution prior to embedding. Four μm sections were cut and stained with haematoxylin and eosin (H&E) and examined microscopically. A lung histopathology scoring system was setup and used to evaluate the severity of the histopathological lesions observed in each animal (Table 1), including lesions affecting the airways and the parenchyma. Six large tissue sections from the left lung were used to evaluate the lung histopathology, three from the cranial lobe and three from the caudal lobe as follows: (a) proximal to the lobar bronchus bifurcation, (b) middle portion and (c) distal to the lobar bronchus bifurcation. Sections were analysed by two qualified veterinary pathologists and peer-reviewed by two other qualified pathologists (veterinary and medical).

In addition, samples were stained using the RNAscope technique to identify the SARS-CoV-2 virus RNA or Interleukin 6 (IL-6) in lung tissue sections. Briefly, tissues were pre-treated with hydrogen peroxide for 10 min (RT), target retrieval for 15 min (98–102 °C) and protease plus for 30 min (40 °C) (Advanced Cell Diagnostics). A V-nCoV2019-S probe (SARS-CoV-2 Spike gene specific), or host species specific IL-6-S probes (Advanced Cell Diagnostics, Bio-techne) were incubated on the tissues for 2 h at 40 °C. In addition, samples were stained using the RNAscope technique to identify the SARS-CoV-2 virus RNA. Amplification of the signal was carried out following the RNAscope protocol using the RNAscope 2.5 HD Detection kit—Red (Advanced Cell Diagnostics, Biotechne). Appropriate positive and negative controls were included in each RNAscope run (Supplementary Fig. 4)

Digital image analysis was performed in RNAscope labelled slides to ascertain the percentage of stained cells within the lesions, by using the Nikon-NIS-Ar package. The presence of viral RNA by ISH was evaluated using the whole lung tissue section slides. For IL-6 mRNA, the areas of histopathological lesions were selected as regions of interest (ROI) and the positively labelled area (red) was calculated by the software after setting the thresholds.

**Viral load quantification by RT-qPCR.** RNA was isolated from nasal wash, throat swabs, EDTA treated whole blood, BAL and tissue samples (nasal cavity, tonsil, trachea and lung). Weighed tissue samples were homogenised and inactivated in RLT (Qiagen) supplemented with 1%(v/v) Beta-mercaptoethanol. Tissue homogenate was then centrifuged through a QIAshredder homogeniser (Qiagen) and supplemented with ethanol as per manufacturer's instructions. Downstream extraction was then performed using the BioSprint™96 One-For-All vet kit (Qiagen) and Kingfisher Flex platform as per manufacturer's instructions. Non-tissue samples were inactivated in AVL (Qiagen) and ethanol, with final extraction using the QIAamp Viral RNA Minikit (Qiagen) as per manufacturer's instructions.

Reverse transcription-quantitative polymerase chain reaction (RT-qPCR) was performed using TaqPath™ 1-Step RT-qPCR Master Mix, CG (Applied Biosystems™), 2019-nCoV CDC RUO Kit (Integrated DNA Technologies) and 7500 Fast Real-Time PCR System (Applied Biosystems™) as previously described[48]. PCR amplicons were quantified against 2019-nCoV_N_Positive Control (Integrated DNA Technologies). Positive samples detected below the lower limit of quantification (LLOQ) of 20 copies/μl were assigned the value of 13 copies/μl, undetected samples were assigned the value of ≤6.2 copies/μl, equivalent to the assays LLOD. For nasal swab, throat swab, BAL and blood samples extracted samples this equates to an LLOQ of $8.57 \times 10^3$ copies/ml and LLOD of $2.66 \times 10^3$ copies/ml.

**Subgenomic RT-qPCR.** Subgenomic RT-qPCR was performed on the Quant-Studio™ 7 Flex Real-Time PCR System using TaqMan™ Fast Virus 1-Step Master Mix (Thermo Fisher Scientific) and oligonucleotides as specified by Wolfel et al.,

with forward primer, probe and reverse primer at a final concentration of 250 nM, 125 nM and 500 nM respectively. Sequences of the sgE primers and probe were: 2019-nCoV_sgE-forward, 5′ CGATCTCTTGTAGATCGTTCTC 3′; 2019-nCoV_sgE-reverse, 5′ ATATTGCAGCAGTACGCACACA 3′; 2019-nCoV_sgE-probe, 5′ FAM- ACACTAGCCATCCTTACTGCGCTTCG-BHQ1 3′. Cycling conditions were 50 °C for 10 min, 95 °C for 2 min, followed by 45 cycles of 95 °C for 10 s and 60 °C for 30 s. RT-qPCR amplicons were quantified against an in vitro transcribed RNA standard of the full-length SARS-CoV-2 E ORF (accession number NC_045512.2) preceded by the UTR leader sequence and putative E gene transcription regulatory sequence described by Wolfel et al. in 2020[49]. Positive samples detected below the lower limit of quantification (LLOQ) were assigned the value of 5 copies/μl, whilst undetected samples were assigned the value of ≤0.9 copies/μl, equivalent to the assays lower limit of detection (LLOD). For nasal swab, throat swab, BAL and blood samples extracted samples this equates to an LLOQ of $1.29 \times 10^4$ copies/mL and LLOD of $1.16 \times 10^3$ copies/mL For tissue samples this equates to an LLOQ of $5.71 \times 10^4$ copies/g and LLOD of $5.14 \times 10^3$ copies/g. Sequences of primers and probes are included in Supplementary Table 1.

**Plaque assay.** Samples were incubated in 24-well plates (Nunc, ThermoFisher Scientific, Loughborough, UK) containing twice washed with Dulbecco's PBS (DPBS) monolayers of Vero E6 cells seeded the previous day at $1.5 \times 10^5$ cells/well under Overlay media consisting of MEM (Life Technologies) containing 1.5% carboxymethylcellulose (Sigma), 4% (v/v) heat-inactivated foetal calf serum (FCS) (Sigma) and 25 mM HEPES buffer (Gibco). After incubation at 37 °C for 120 h, they were fixed overnight with 10% (w/v) formalin/PBS, washed with tap water and stained with methyl crystal violet solution (0.2% v/v in 40% (v/v) Ethanol) (Sigma).

**Plaque reduction neutralisation test.** Neutralising virus titres were measured in heat-inactivated (56 °C for 30 min) serum samples. SARS-CoV-2 was diluted to a concentration of $1.4 \times 10^3$ pfu/ml (70 pfu/50 μl) and mixed 50:50 in 1% FCS/MEM with doubling serum dilutions from 1:10 to 1:320 in a 96-well V-bottomed plate. The plate was incubated at 37 °C in a humidified box for 1 h to allow the antibody in the serum samples to neutralise the virus. The neutralised virus was transferred into the wells of a washed plaque assay 24-well plate (see plaque assay method), allowed to adsorb at 37 °C for a further hour, and overlaid with plaque assay overlay media. After 5 days incubation at 37 °C in a humified box, the plates were fixed, and plaques counted.

A mid-point probit analysis was used to determine the dilution of antibody required to reduce SARS-CoV-2 viral plaques by 50% (PRNT50) compared with the virus only control (n = 5). Analysis was conducted in R[50] and the script was based on a source script from Johnson et al. [51].

**ELISA.** A full-length trimeric and stabilised version of the SARS-CoV-2 Spike protein (amino acids 1-1280, GenBank: MN MN908947) was developed and kindly provided by Florian Krammer's lab as previously described[52]. Recombinant SARS-CoV-2 Receptor-Binding-Domain (319-541) Myc-His was developed and kindly provided by MassBiologics, USA. Recombinant SARS-CoV-2 Nucleocapsid phosphoprotein (GenBank: MN908947, isolate Wuhan-Hu-1) was expressed and purified from Escherichia coli as full-length nucleoprotein (amino acids 1-419) with a C-terminal 6xHis-Tag (REC31812-100, Batch #20042310, Native Antigen Company).

Spike-, Spike RBD- and NP-specific IgG responses were determined by ELISA. High-binding 96-well plates (Nunc Maxisorp) were coated with 50 μl per well of 2 μg/ml Spike trimer, Spike RBD or NP in 1X PBS (Gibco) and incubated overnight at 4 °C. The ELISA plates were washed five times with wash buffer (1 X PBS/0.05% Tween 20 (Sigma)) and blocked with 100 μl/well 5% FBS (Sigma) in 1 X PBS/0.1% Tween 20 for 1 h at room temperature. After washing, serum samples previously 0.5% Triton-inactivated were serially diluted in 10% FBS in 1 X PBS/0.1% Tween 20, 50 μl/well of each dilution were added to the antigen coated plate and incubated for 2 h at room temperature. Following washing, anti-monkey IgG conjugated to HRP (Invitrogen) were diluted (1:10,000) in 10% FBS in 1 X PBS/0.1% Tween 20 and 100 μl/well were added to the plate, then incubated for 1 h at room temperature. After washing, 1 mg/ml O-phenylenediamine dihydrochloride solution (Sigma) was prepared and 100 μl per well were added. The development was stopped with 50 μl per well 1 M Hydrochloric acid (Fisher Chemical, J/4320/15) and the absorbance at 490 nm was read using Softmax 7.0. Endpoint titres and statistical analyses (Kruskal–Wallis one-way ANOVA) were performed with Graph Pad Prism 8.0. The cut-off was set at the average Optical Density of samples collected from naïve animals (Day 0) + 3 Standard Deviation.

**Mononuclear cell Isolation.** PBMCs were isolated from whole blood anticoagulated with heparin (132 Units per 8 ml blood) (BD Biosciences, Oxford, UK) using standard methods. Of note is that the material used for density gradient centrifugation was adjusted dependent on the macaque species, with a Ficoll Histopaque gradient (GE Healthcare, USA) used with rhesus macaque blood and a Percoll gradient (GE Healthcare) used with cynomolgus macaques. Mononuclear cells (MNC) were isolated from spleen and lung tissue samples using an Octo-MACS tissue dissociation device (Miltenyi Biotec). Lung tissue samples were dissected into approximately 5mm³ pieces and incubated for 1 h in a solution of 772.8

**Table 1 Pulmonary histopathology scoring system.**

| Lesion | Score 0 (normal) | Score 1 (minimal) | Score 2 (mild) | Score 3 (moderate) | Score 4 (severe) |
|---|---|---|---|---|---|
| Bronchial epithelial degeneration/necrosis with presence of exudates and/or inflammatory cell infiltration | None | Occasional (1 or 2) bronchi affected. | Present in multiple airways; up to 25% of bronchi affected | Present in multiple airways; between 26–50% of bronchi affected | Present in multiple airways; over 50% of bronchi affected |
| Bronchiolar (primarily terminal) epithelial degeneration/necrosis with presence of exudates and/or inflammatory cell infiltration | None | Occasional (1 or 2) bronchioli affected | Present in multiple airways; up to 25% of bronchioli affected | Present in multiple airways; between 26–50% of bronchioli affected | Present in multiple airways; over 50% of bronchioli affected |
| Perivascular inflammatory infiltrates (cuffing) | None | Occasional incomplete, or loosely formed cuffs | Numerous cuffs; predominantly incomplete and loosely formed with lesser well-formed complete cuffs | Numerous cuffs; approximately half or more well-formed, and may have few broad, dense cuffs | Numerous cuffs; predominantly well-formed with numerous broad, dense cuffs |
| Peribronchiolar inflammatory infiltrates (cuffing) | None | Occasional incomplete, or loosely formed cuffs | Numerous cuffs; predominantly incomplete and loosely formed with lesser well-formed complete cuffs | Numerous cuffs; approximately half or more well-formed, and may have few broad, dense cuffs | Numerous cuffs; predominantly well-formed with numerous broad, dense cuffs |
| Acute diffuse alveolar damage (necrosis of pneumocytes) | None | Small numbers of foci; up to 5% of slide affected | Multiple foci; between 6–25% of the slide affected | Increased numbers of foci; between 26–50% of the slide affected | Numerous foci; over 50% of the slide affected |
| Alveolar cellular exudate and oedema and/or fibrin | None (alveolar macrophages at physiological levels) | Occasional alveoli; up to 5% of slide affected | Confluent alveoli; between 6–25% of the slide affected | Confluent alveoli; between 26–50% of the slide affected | Confluent alveoli; affecting over 50% of the slide |
| Alveolar septal inflammatory cells and cellularity | Normal septae; typically 1-2 (occasionally 3) nucleated cells wide; absence of inflammatory cells | Thickening of the alveolar walls by inflammatory cells; up to 5% of the slide affected | Thickening of the alveolar walls by inflammatory cells; between 6–25% of the slide affected | Thickening of the alveolar walls by inflammatory cells; between 26–50% of the slide affected | Thickening of the alveolar walls by inflammatory cells; over 50% of the slide affected |

U/ml collagenase + 426 U/ml DNase (both from Sigma) diluted in Earle's balanced salt solution supplemented with 200 mg/ml Calcium Chloride (Gibco, Life Technologies, Renfrew, UK), at 37 °C with continual gentle mixing of the tube. The homogenised solution was passed through a 70 μm cell filter (BD Biosciences) and the mononuclear cells separated by Ficoll Histopaque density gradient centrifugation. PBMCs and MNC isolated from tissues were stored at −180 °C until resuscitated for analysis.

**Resuscitation of cryopreserved cells**. PBMCs and MNC were thawed, washed in R10 medium (consisting of RPMI 1640 supplemented with 2 mM L-glutamine, 50 U/ml penicillin- 50 μg/ml streptomycin, and 10% heat-inactivated FBS) with 1 U/ml of DNase (Sigma), and resuspended in R10 medium and incubated at 37 °C 5% $CO_2$ overnight.

**ELISPOT**. An IFNγ ELISpot assay was used to estimate the frequency and IFNγ production capacity of SARS-CoV-2-specific T-cells in PBMCs using a human/simian IFNγ kit (MabTech, Nacka. Sweden), as described previously[53]. The cells were assayed at $2 \times 10^5$ cells per well, unless there were not enough cells, in which case $1 \times 10^5$ cells were used. Cells were stimulated overnight with SARS-CoV-2 peptide pools and 'megapools' of the spike protein (Mimotopes, Australia). Ten peptide pools were used, comprising of 15mer peptides, overlapping by 11 amino acids and offset by 4 amino acids. Peptides were resuspended in 10% DMSO to a median concentration of 9.22 mg/ml. Peptides were then pooled and aliquoted. Before use peptides were resuspended to a concentration of 5 μg/peptide in media, they were then loaded onto the ELISpot plate, 50 μl/well. This peptide library is based on GenBank: MN908947.3 sequence. Full sequences of peptides are included in Supplementary Table 2. The three megapools were made up as such: Megapool 1 (MP1) comprised peptide pools 1-3, Megapool 2 (MP2) comprised peptide pools 4-6 and Megapool 3 (MP3) comprised of peptide pools 7-10. All peptides were used at a concentration of 1.7 μg per well. Phorbol 12-myristate (Sigma) (100 ng/ml) and ionomycin (CN Biosciences, Nottingham, UK) (1 mg/ml) were used as a positive control. Results were calculated to report as spot forming units (SFU) per million

cells. All SARS-CoV-2 peptides and megapools were assayed in duplicate and media only wells subtracted to give the antigen-specific SFU. ELISPOT plates were analysed using the CTL scanner and software (CTL, Germany) and further analysis carried out using GraphPad Prism (version 8.0.1) (GraphPad Software, USA) Immunophenotyping and Intracellular cytokine staining assays. Naïve animal samples were taken from previous studies.

Intracellular cytokine staining (ICS) and immunophenotyping assays were performed using $1 \times 10^6$ PBMC or MNC in R10 medium (described above). For intracellular cytokine staining, these cells were stimulated with a 10 μg/ml solution of CD28 and CD49d co-stimulatory antibodies (both from BD Biosciences) and 1ug/ml of 15-mer overlapping peptide pools spanning either the SARS-CoV-2 spike (S), nucleocapsid (N) or membrane glycoprotein (M) sequence (Miltenyi Biotec, Bisley, UK) or 5 μg/ml staphylococcal enterotoxin b (SEB) (Sigma), or R10 medium with matched concentration of DMSO as negative control, for a total of 16 h at 37 °C, in a 5% $CO_2$ supplemented incubator. Anti-CD107a-AF488 (BD Biosciences) was included during cell stimulations. Following the initial 2 h of incubation, the protein transport inhibitor Brefeldin-A (Sigma) was added at a final concentration of 10 μg/ml. Following incubation, cells were washed with FACS buffer consisting of PBS + 1% FCS and incubated for 30 min at room temperature with optimal dilutions of the amine-reactive Live/Dead Fixable Red viability cell stain (Life Technologies) and the antibodies CD4 PerCP-Cy5.5, CD8 APC-Fire750, CD69-BV510, (all from BD Biosciences) and CD20- Pe-Dazzle-594, γδ-TCR-BV421 (Biolegend, London, UK) prepared in BD Biosciences Brilliant stain buffer (BD Biosciences, Oxford, UK). Following surface marker staining, the cells were washed and then permeabilised by incubation at room temperature for 15 min with Fix/Perm reagent (BD Biosciences) before washing with Permwash buffer (BD Biosciences). Intracellular antigen staining was applied by incubation at room temperature for 30 min with the antibodies CD3-AF700, IFN-γ-PeCy7, TNF-α-BUV395, GM-SCF-PE (all from BD Biosciences, Oxford, United Kingdom), IL-2-APC (Miltenyi Biotech Ltd), IL-17-BV711 (Biolegend, London, UK) prepared in brilliant stain buffer. For immunophenotyping assays, cells were washed with FACs buffer by centrifugation before staining with amine-reactive Live/Dead Fixable violet viability cell stain as per the manufacturer's instructions (Life Technologies).

Cells were then incubated for 30 min at room temperature with optimal dilutions of the following antibodies: anti-CD4-PerCP-Cy5.5, anti-CD8-APC-Fire750 anti-CD11c-PE. anti-CD14-APC, anti-CD16-BV786, anti-CD20-PE-Dazzle (all from BioLegend); anti-CD3-AF700, anti-CD56-BV605, anti-HLA-DR-BUV395 (all from BD Biosciences); anti-CD159a-PC7 (Beckman Coulter) prepared in brilliant stain buffer. BD Compbeads (BD Biosciences) were labelled with the above fluorochromes for use as compensation controls. Following antibody labelling, cells and beads were washed by centrifugation and fixed in 4% paraformaldehyde solution (Sigma) prior to flow cytometric acquisition.

**Flow cytometric acquisition and analysis.** Cells were analysed using a five laser LSRII Fortessa instrument (BD Biosciences) and data were analysed using FlowJo (version 9.7.6, BD Biosciences). Cytokine-producing T-cells were identified using a forward scatter-height (FSC-H) versus side scatter-area (SSC-A) dot plot to identify the lymphocyte population, to which appropriate gating strategies were applied to exclude doublet events, non-viable cells and B cells (CD20+). For ICS analysis, sequential gating through CD3+, followed by CD4+ or CD8+ gates were used before individual cytokine gates to identify IFN-γ, IL-2, TNF-α, GM-CSF and IL-17, CD107a and CD69 stained populations. In immunophenotyping datasets, classical-, non-classical-monocytes and monocyte derived dendritic cells (mDCs) were identified by FSC and SSC characteristics and by the expression pattern of HLA-DR, CD14, CD16 and CD11c within the live CD3-, CD20- population. Similarly, natural killer cells subsets were identified by expression of CD8, CD159a, CD56 and CD16 within live CD3- lymphocyte subsets. Polyfunctional cells were identified using Boolean gating combinations of individual cytokine-producing CD4 or CD8 T-cells. The software package PESTLE version 1.7 (Mario Roederer, Vaccine Research Centre, NIAID, NIH) was used for background subtraction to obtain antigen-specific polyfunctional ICS cytokine responses, Graphpad Prism (version 8.0.1) was used to generate graphical representations of flow cytometry data. Gating strategy for flow cytometry analyses are summarised in Supplementary Figs. 5, 6.

**Sequencing, data collection and data deposition.** Total RNA was extracted from whole blood with the QIAmp viral RNA extraction kit and eluted in pure water. Following the manufacture's protocols, total RNA was used as input material into the QIAseq FastSelect–rRNA HMR (Qiagen) protocol to remove cytoplasmic and mitochondrial rRNA with a fragmentation time of 7 or 15 min. Subsequently, the NEBNext® Ultra™ II Directional RNA Library Prep Kit for Illumina® (New England Biolabs) was used to generate the RNA libraries, followed by 11 or 13 cycles of amplification and purification using AMPure XP beads. Each library was quantified using Qubit and the size distribution assessed using the Agilent 2100 Bioanalyser and the final libraries were pooled in equimolar ratios. The raw fastq files (2 ×150 bp) generated by an Illumina® NovaSeq 6000 (Illumina®, San Diego, USA) were trimmed to remove Illumina adapter sequences using Cutadapt v1.2.1[54]. The option "−O 3" was set, so the that 3' end of any reads which matched the adapter sequence with greater than 3 bp was trimmed off. The reads were further trimmed to remove low quality bases, using Sickle v1.200[55] with a minimum window quality score of 20. After trimming, reads shorter than 10 bp were removed. Sequence reads that mapped to *Macaca mulatta* or *Macaca fascicularis* are deposited under NCBI bioproject: PRJNA681111.

**Bioinformatic Analysis.** Hisat2 v2.1.0[56] was used to map the trimmed reads on the *Macaca mulatta* or *Macaca fascicularis* reference host genome assemblies (release-94) downloaded from the Ensembl FTP site. The resultant Sam files were processed by featureCounts v2.0.0[57] with the default setting to generate raw read counts per gene. One-to-one orthologs were defined as the orthologous genes of *M. mulatta* and *M. fascicularis* in Ensembl (release-94) with reciprocally highest homologue ortholog confidence. For comparison between the two species, 24,300 one-to-one orthologs with their raw read counts were used for analysis. For the gene-network analysis on the blood samples, first low-expression genes (at least 1 read per million in 1 of the samples) were filtered and gene counts in the *Macaca sp.* datasets were normalised and log-transformed with the command rpkm() in EdgeR[58]. Then 25 Consensus modules in *M.mulatta* and *M. fascicularis* were derived from WGCNA[59] with blockwiseConsensusModules function (power = 8, TOMType = "signed", corType = "bicor", maxPOutliers=0.05, minModuleSize=30, deepSplit = 4). Species specific modules were identified by relating consensus modules to the *M. mulatta* modules and *M. fascicularis* modules. The GO enrichment analysis for differentially expressed genes and modules were performed using clusterProfiler[60]. The module function was identified by dual enrichment.

**Reporting Summary.** Further information on research design is available in the Nature Research Reporting Summary linked to this article.

## Data availability
All data and materials used in the analysis are presented in the main manuscript and supplementary information files. Accession number PRJNA681111 is available at https://www.ncbi.nlm.nih.gov/bioproject/PRJNA681111 Source data are provided with this paper.

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

## Acknowledgements
The authors would like to thank J. Druce and M.G. Catton from the Victorian Infectious Diseases Reference Laboratory, Royal Melbourne Hospital, At the Peter Doherty Institute for Infection and Immunity, Victoria, 3000, Australia, for providing the SARS-CoV-2 isolate used in this study. This work was funded by the Coalition of Epidemic Preparedness Innovations (CEPI) and the Medical Research Council (Project CV220-060, "Development of an NHP model of infection and ADE with COVID-19 (SARS-CoV-2). The bioinformatics analysis was part supported by the US FDA contract number 75F40120C00085 'Characterisation of severe coronavirus infection in humans and model systems for medical countermeasure development and evaluation' awarded to JAH and MWC.

## Author contributions
F.J.S., K.A.R., Y.H., J.A.T., S.G.F., K.R.Bw., S.S., M.J.D., S.C., B.H. and M.W.C. conceived the study. F.J.S., L.H., C.L.K., G.P., E.L.R. and A.G.N. performed the pathological analyses. A.D.W., L.S., C.S., S.Lo., O.D.-P., K.G., A.M., A.L.M. and T.T. performed the immunological studies. KRBw grew viral stock and optimised virology techniques. S.A.F., P.B., B.E.C., R.C., D.J.H., T.H., C.M.K.H., V.L., D.N., J.P., I.T., S.T., N.R.W. and M.J.D. performed animal procedures and sampling processing. G.S.S., K.E.G., H.E.H., R.J.W., L.A., E.B., K.R.Bu., M.A., N.S.C., D.K., K.J.G., J.G., R.H., S.Le., E.J.P., S.P., C.T. and N.W. performed virological and molecular biology techniques. MJE performed statistical analyses. F.V.G. devised the scan protocols, the scoring system and reported the CT scans. I.G.-D., E.V., C.N. and A.D. performed the sequencing analysis. X.D. and J.A.H. performed the bioinformatic data analysis. F.J.S., A.D.W., M.J.D., S.S., J.A.H. and M.W.C. wrote the manuscript that was reviewed and accepted by all authors before submission.

## Competing interests
The authors declare no competing interests.
