## [Peer Review File · Nature Communications]

Reviewers' Comments:

Reviewer #1:

Remarks to the Author:

General Comments:

The overall objective of this study is to compare the Rhesus and Cynomolgus macaque response to COVID-19 infection. Because Rhesus macaques have been the primary nonhuman primate animal model used to investigate COVID-19 pathology and therapeutics, the authors proposed to evaluate the suitability of Cynomolgus macaques as an alternative nonhuman primate animal model. The authors further rationalize that possible Rhesus macaque shortages from ongoing studies may necessitate the use of different nonhuman primate species for future COVID-19 studies.

To address this gap in knowledge, the authors comparably infected and assessed 6 Indian Rhesus macaques with 6 Mauritian Cynomolgus macaques. Animal groups were balanced for sex, age, and body weight. All animals received identical dosing regimens for virus. The authors evaluated viral replication, histopathology, neutralizing antibodies, IgG antibody levels, and cellular immune phenotypes (IFN gamma production/FACS analysis).

Both species of macaques developed a mild disease course as a result of SARS-CoV-2 infection; there were no apparent differences detected from the measured parameters. Viral replication and histopathology of the lung appeared similar between the two monkey species assessed, with slightly higher (but apparently not significant) viral RNA/IL-6 in Cynomolgus macaques at the early timepoint of assessment. Some differences in immune profiles were detected, with Cynomolgus macaques appearing to generate more robust IFN gamma responses.

Overall, no statistically significant differences were detected between the two species, however many of the end points had a small sample size. At a minimum, the authors should comment on the limitation of having a small sample size in the discussion.

Major Comments:

1. A schematic with a timeline for various assessments would be helpful.
2. For Figure 1, the text in the body of the manuscript indicates that 2 Cynos and 2 Rhesus were evaluated by CT. However, the Figure only shows 2 Cynos and 1 Rhesus. What is the basis of A and A'? How were these fields selected? What is the rationale for evaluating at 18 days post infection? Why were these particular animals selected for CT analysis? It would also have been helpful to include a CT from naive animals for comparison.
3. It appears that there is low level virus detected in the upper airways almost 3 weeks post challenge. Is this the most prolonged period of assessment in nonhuman primates?
4. The authors indicate that viable virus (as tested by plaque assay) could not be detected after 4 days post challenge, but viral RNA is detectable for 2 weeks following this period suggesting continued replication at this site. What could explain this discordance?
5. Controls and associated images should be provided for in situ hybridization data.
6. There appears to be evidence of baseline neutralizing antibody in at least two of the Cynomolgus macaques. How did these animals in particular respond to infection? Some of the Cynomolgus macaques also appear to have much higher specific IgG responses for days 8-9. Are the highly responsive animals the same animals with elevated neutralizing antibodies at baseline? Is this evidence of a memory response (previous infection)?
7. Although it is understood that sample size is small, it would be of interest to comment on whether sex associated with any parameters in the study.

Reviewer #2:

Remarks to the Author:

In this report, Salguero et al. compare SARS-CoV2 infection in rhesus macaques (RM) and cynomolgus macaques (CM) as animal models for studying COVID-19. This is important as there is a scarcity for the availability of rhesus macaques and the availability of additional species of non-human primates (NHPs) as an animal model will help to accelerate the development of immunotherapies and vaccines for COVID19. To accomplish the goals, the authors infected 6 CMs and 6 RMs with SARS-CoV-2 Victoria/01/2020 virus and performed viral RNA measurements and CT scans, antibody responses and T cell responses. Most of these analyses are thorough. The analyses revealed that both species show comparable levels of virological and immunological findings. The authors should address the following comments/concerns.

1) Fig. 1 needs proper legend.

2) Identify groups in Fig 2.

3) The authors should write a sentence in the methods section about peptide pools covering which region of the protein. For example PP1 covers NTD of Spike protein, PP2 covers....!

4) Provide a reference for the following sentence; Histopathological changes were comparable in both animal species, and closely resembled that seen in human cases following a mild/moderate clinical course (line 422-423).

Reviewer #3:

Remarks to the Author:

Salguero and colleagues provide data on a comparative challenge experiment with SARS-CoV2 in Asian macaques where the primary variable would be the species of macaque used (rhesus vs cynomolgus). The study involves a small cohort (n=6 per species) of sex balanced, pubescent animals (2-4 years old, puberty in these species occurs between 2-5 years of age). After intranasal/IT challenge, the animals were monitored and sampled longitudinally by various methods. Scheduled euthanasia events were performed to track pathological changes in tissues on subsets of animals over time (n=2, sex balanced). The central claim of the work is that cynomolgus macaques (of Mauritian origin) challenged with SARS-CoV2 result in similar responses to challenge to that of rhesus macaques (Indian origin).

The manuscript needs a fair amount of work to clear up a number of serious issues.

1. Manuscript title: I would seriously blunt the claim of either of these models being "an authentic model of COVID19". What is described here is an infection model at best.

2. This reviewer agrees with the urgency to find alternative non-human primate models (NHP) that accurately recapitulate human COVID19 as the current trade blockade of this precious resource from Eastern Asia has no doubt put a serious strain on NHP resources across the globe. While the work holds some novelty as a limited comparison between two species and does demonstrate value of the Mauritian cynomolgus macaque as an infection model of SARS-CoV2, which much of the field would agree is comparable to what is known about rhesus as a model. The major caveat here is that cynomolgus macaques, regardless of origin, are almost as hard to source as rhesus currently. So authors claim that using cynomolgus as a replacement does not hold the perceived value claimed by the authors, at least in the eyes of this reviewer.

3. A number of fundamental misconceptions are listed throughout the text that make it clear that the authors are not completely in tune with the state of the field. In particular, the authors assertion that a consensus that rhesus macaques are the "optimal study species" is unfounded (Abstract and lines 75-77). While it is true that early in the current pandemic a number of studies have used the model to assess vaccine candidates and development of immunity, these were largely done before more complete evaluations in other species were completed. The authors will no doubt realize that a number of investigators have independently validated the African Green Model as an equivalent, if not superior, model of human COVID19 (Hartman et al PLOS Pathogens 16(9): e1008903).

<https://doi.org/10.1371/journal.ppat.1008903>, Woolsey et al, bioRxiv 2020.05.17.100289; doi: <https://doi.org/10.1101/2020.05.17.100289>, Blair et al, bioRxiv 2020.06.18.157933; doi: <https://doi.org/10.1101/2020.06.18.157933> ; Cross et al., Virol J 17, 125 (2020). <https://doi.org/10.1186/s12985-020-01396-w> , Johnston et al, bioRxiv 2020.06.26.174128; doi: <https://doi.org/10.1101/2020.06.26.174128>). The authors also assert that for SARS-CoV the preferred species are rhesus (LINE 66-69), which again is arguably false as the African Green Monkey model has clearly been demonstrated (in parallel) to be equivalent if not superior to both Asian macaque species, especially rhesus (McAuliffe et al Virology. 2004 Dec 5; 330(1): 8-15, Smits et al., Journal of Virology Apr 2011, 85 (9) 4234-4245; DOI: 10.1128/JVI.02395-10 and other...). There is also growing understanding in the field that the rhesus is more of a viral clearance model, than one truly reflective of global pathogenesis reflective of human COVID19. While infection models are certainly valuable for demonstrating immune responses, without demonstration of associated pathology, their value is limited in terms of detailed pathological studies or even evaluation of predictive efficacy of therapeutics (especially post exposure).

4. BAL collection: This is not a standard way of collecting BALs as reported in all other NHP SARS-CoV2 studies, this is more of an ex-vivo flushing of the lung lobe(s?). Given the animals are euthanized via intracardiac injection, there is major suspect to the validity of any data gleaned from these samples due to potential artifact associated with cellular contact with the euthanasia solution. The data for RNA is likely good, but I'm not sure I would be confident in any cellular immunological or histological analysis done on lungs sampled this way.

5. Nasal washes: Why not just swab? This is what the majority of the field is doing and is more reflective of human sampling. Nasal washes are typically done on smaller animal models such as ferrets or hamsters as swabbing is not practical approach. The approach makes comparability with other studies challenging.

6. Line 151: This is high? Was virus isolated from these samples? If high, there should have been no problem. Between 6-7 logs of virus is just about the threshold of where one could expect to have any success isolating infectious virus from a biological sample in such a way that it could be effectively quantified by plaque assay. I would temper the claim.

7. Line 170 and throughout. Virus was only isolated (at very low levels) 1-3 dpc, how do you know this was not just left over inoculum from the challenge?

8. Line 386, not the first time: Johnston et al, bioRxiv 2020.06.26.174128; doi: <https://doi.org/10.1101/2020.06.26.174128>

9. There is mention of credentials of the radiologist as being an "expert". What does that mean? Related to this, are the pathologists equally qualified? This would be important justification for trying to develop a "new method for lung histopathology scoring". Abstract, line 42.

10. Study terminology: The description is somewhat confusing, what does 4/5 and 14/15 etc mean? Did the authors take body fluids on the first day (4) and then culled on the second day (5)? Introduction, lines 91-93.

11. Why was CT only done on limited animals and only on day 18, why not before when the virus is running rampant in the lungs? With no kinetics of changes over time, these data hold little value. Results, lines 114-115.

12. The only thing noted on CT was ground glass opacity but later in the text the authors mention they say that "Features characteristic of COVID-19 in human patients, such as ground glass opacity, consolidation and crazy paving were identified on CT scans collected from both SARS-CoV-2 challenged rhesus and cynomolgus macaques in line with..." Discussion, line 401-404 vs Results, lines

120-125. Reconciliation needed.

13. "Gross pathological changes were found in the lungs of all animals....", so where are the gross images? How about a gross score to determine severity and to correlate with the histopath scoring?....Pathological changes, lines 176-181.

14. In regards to tissue handling, were the lungs sufflated? were all lung lobes sampled? I don't think so because the M&M stats this... "...BAL was collected at necropsy from the right lung. The left lung was dissected prior to BAL collection and used for subsequent histopathology and virology procedures..." I think there are inherent errors if not all lung lobes are collected and evaluated for this disease since it's so multifocal and mild so if trying to develop a method for scoring we should only look at half the lung? Also if lobes are not sufflated it is really difficult from animal to animal to determine alveolar thickness due to inflammatory infiltrates vs just collapse with multiple alveolar walls side by side. Post-mortem examination and histopathology, lines 610-614.

15. Along the lines of #8, how many sections of lung were examined and from where? Most seasoned pathologists working with respiratory viruses make a big deal about sampling the hilar portion and the distal portion. 3 per lobe would be considered standard practice. It is stated that the authors take 3 samples, but they do not define where the samples are taken from... "Three tissues sections from each left lung lobe were used to evaluate the lung histopathology." Also did they only define the cranial and caudal portions? or did they take the cranial (upper lobe) and caudal portions (middle lobe) of the cranial lobe and the caudal lobe (Lower lobe) so we are not talking looking at 6 slides vs 9 slides vs the 18 slides that we look at, there always is variability when slides are reviewed, so a more clear understanding of the pathologists process in determining confidence in their scoring would be particularly helpful to build confidence in the proposed scoring method. Post-mortem examination and histopathology, lines 625-626.

16. There is a lot of heavy use buzz word of DAD. There are some specific requirements for you to hang your hat on the diagnosis of DAD, and the data as provided don't seem to back this up. DAD means diffuse alveolar damage. It's hugely important in human COVID-19 but the lesions really are not that impressive in all of the animal models (with the exception of the Mink--there is some beautiful lesions in mink). The first thing that qualifies DAD is the presence of hyaline membranes, this paper fails to point out any hyaline membranes by IHC. Its likely however, that the authors may be challenged to observe this feature without proper sufflation of the lungs during tissue preparation. I think the authors could conclude there was alveolar damage but not DAD unless they provide the hyaline membranes and the severity of the lesions (aka it needs to be diffuse (the whole lung) otherwise it's focally diffuse. "Overall, diffuse alveolar damage (DAD) was a prominent feature in the affected areas (see it's focally), characterized by individual, shrunken, eosinophilic cells in alveolar walls with pyknotic or karyorrhectic nuclei" Pathological changes, lines 189-191.

17. More path issues to address...

a. "in these areas, alveolar spaces were often obliterated by collapse of the thickened and damaged alveolar walls..." If the lungs were not sufflated they are expected to collapse when the air escapes at opening the chest cavity/removal of the pluck. Pathological changes, line 192.

b. "...admixed with fibrin, polymorph neutrophils (PMNs), enlarged alveolar macrophages and other round cells (possibly detached type 2 pneumocytes)." Ok, PMN stands for polymorphonuclear cells not polymorph neutrophil.also PMN includes all granulocytes so, so may be best to just say neutrophils here; Pathological changes, lines 197-198.

c. "enlarged alveolar macrophages" how do you know they are enlarged? Typically they are described as activated or reactive i don't think it's proper to say enlarged (although it does get the point across) and if either of these its suggested to demonstrate activation or reactivity with a specific marker. Pathological changes, lines 197-198.

d. "other rounds cells" this means leukocytes like lymphocytes, plasma cells etc...a pneumocyte is not a round cells it's epithelial and you can easily run an IHC marker to determine (as many others in

the field have) so consider running a pancytokeratin or surfactant IHC to be sure and not to guess. Also I don't think it's proper to say type 2 it's always noted as type II in the roman numbers. Pathological changes, lines 197-198.

e. "In distal bronchioles and bronchiolo-alveolar junctions, degeneration and sloughing of epithelial cells was present, with areas of attenuation and foci of plump, type 2 pneumocytes representing regeneration." This could also be bronchiolization so the epithelium is coming from the bronchiolar epithelium and not just the type II pneumocytes that are in the alveolar walls. Need to define where you see this. Type II pneumocytes hyperplasia will likely not be in the distal bronchioles but more likely in the alveolar junctions. Pathological changes, lines 198-201.

f. PMN issue again, just need to put neutrophils. Continuing to describe these as PMN confuses the issue with generalized granulocytes. Also, is this inflammation associated with the previously described epithelial sloughing? otherwise the sloughing maybe a postmortem change and not a pathologic change. Pathological changes, line 206.

g. "In the lumen of some airways, fibrillar, eosinophilic material (mucus)...." Mucus is always slightly basophilic or amphophilic...eosinophilic is edema or fibrin. Also mucus is not fibrillar that is really pointing towards fibrin...so its suggested to run a fibrin stain (PTAH or IHC). Pathological changes, lines 207-208.

h. "...resembling syncytial cells..." please don't guess, run IHC for epithelial cells or macrophage marker or just leave it at multinucleated cells. Pathological changes, lines 210-211.

i. "...and hyperplasia of alveolar macrophages..." this is a strange term, standard terminology is to say increased numbers of alveolar macrophages because they are recruited there rather than generated there. Pathological changes, line 217.

j. "Positive cells were also observed rarely..." What cells are they, mononuclear cells? Pathological changes, line 225.

k. "...expressing IL-6 mRNA..." Not an overly convincing way to say that IL-6 was associated with the lung tissue inflammation in any meaningful way without evidence at the protein level and perhaps more convincing association with inflammation and viral antigen (or even RNA). Also no cytokine analysis at the protein level to support actual generation of the marker rather than a few cells lighting up with the marker? Transcription does not always equal translation... Pathological changes, line 227 also listed in Discussion, lines 435-439.

l. "...rhesus macaques within the alveolar walls and interalveolar septa..." whats the difference?...alveolar walls and interalveolar septa? Unless the authors mean pneumocytes (as the alveolar walls) and capillaries with macrophages (as the interalveolar septa)? This seems very poorly defined and unfounded. Will require better descriptions Pathological changes, line 231.

18. "In summary, using the histopathology scoring system developed here, the scores..." Where are the definitions of the scoring done? As provided, this reviewer cannot get a clear view understanding that the paper was written using the criteria in the table provided. I think at best that can be said here is how they graded the left lung lobes when comparing rhesus and cynos in this paper. I don't think this will be a new standard of scoring, also any pathologist will be looking at these areas and i think it's only applicable to the early stages of the disease, what about fibrosis later in the disease? As no hyaline membranes represented as part of the scoring, the value of the new method is further reduced as this feature is extremely important in humans and is good to know if they are present in animal models. Also do they score this per animal or per lung lobe or per slide examined? What is done if there is not a bronchus in your section? How do you include IHC or ISH scoring? The scoring is on a percent basis, so how do you assess that? Look at 10 high power fields and average or you count the whole slide? What if the tissue sample is minimal? Lots of questions in regards to the robustness of this scoring system but do think these are valid lesions to look at and consider when reading out the slides, just not sure a score is needed for each of these. Once you get a score how does that correlate to clinical or blood pathologies? Pathological changes, line 251.

13) Refine the summary paragraphs, I think just needs to be reworded to tie everything together, seems a little thrown together. Pathological changes, lines 251-273.

14) I don't know what lung MNC samples are, i either missed it or it's not defined. Composition and functional profile of the cell mediated immune response, line 325.

15) "...mild/moderate clinical course. Typical changes of ARDS were observed, with ..." ARDS is not a mild/moderate clinical course (that i'm aware of, am i wrong?) I think they can say something like focal (diffuse) alveolar damage, pneumonia, type II pneumocyte hyperplasia, etc were noted...additionally they go on to say, "these histopathological lesions are compatible to those observed in human patients..." they aren't because nobody is getting biopsies (Tom and I had to search hard for non fatal histology..there's like 2 papers out there)...the human publications are largely only fatal cases and the NHPs are not representative of that.....Discussion, line 424-428

16. Only two animals per timepoint per species were sampled. The animal to animal variability is very high with this virus, there is no way to conclude statistical significance on such a limited number of samples.

FIGURE LEGENDS

Figure 1: The CT is not very sharp, I also don't know what A vs A' is, Is this a positional demarcation?

Figure 3: B: No hyaline membranes shown with the DAD, If it's true DAD there should be some hyaline membranes.... should be type II pneumocyte hyperplasia not just pneumocyte hyperplasia and again please run IHC for epithelial cells, don't guess for the multinucleated cells

D: DAD? Ok but again no hyaline membranes

F: alveolar macrophage hyperplasia? that's a strange term it's increased alveolar macrophages.

G: bronchial exudates? I don't see this here in the image.

I: parenchymal collapse (it should if you didn't sufflate the lungs)

J: perivascular cuffing? I can't tell if this is a vessel or not need closer image also more of a cuff if possible

K: bronchiole regeneration? I'm not sure why they say this here i can't see other than epithelium lining a bronchiole so maybe it's piling up?

Figure 4: Confusing figure. Would like to just see the histology images of the lesions and the lack of lesions at the later time points. I don't really see the big difference in the histology, maybe it need to be presented differently?

NCOMMS-20-35829 Response to Reviewers

Reviewer #1 (Remarks to the Author):

General Comments:

The overall objective of this study is to compare the Rhesus and Cynomolgus macaque response to COVID-19 infection. Because Rhesus macaques have been the primary nonhuman primate animal model used to investigate COVID-19 pathology and therapeutics, the authors proposed to evaluate the suitability of Cynomolgus macaques as an alternative nonhuman primate animal model. The authors further rationalize that possible Rhesus macaque shortages from ongoing studies may necessitate the use of different nonhuman primate species for future COVID-19 studies.

To address this gap in knowledge, the authors comparably infected and assessed 6 Indian Rhesus macaques with 6 Mauritian Cynomolgus macaques. Animal groups were balanced for sex, age, and body weight. All animals received identical dosing regimens for virus. The authors evaluated viral replication, histopathology, neutralizing antibodies, IgG antibody levels, and cellular immune phenotypes (IFN gamma production/FACS analysis).

Both species of macaques developed a mild disease course as a result of SARS-CoV-2 infection; there were no apparent differences detected from the measured parameters. Viral replication and histopathology of the lung appeared similar between the two monkey species assessed, with slightly higher (but apparently not significant) viral RNA/IL-6 in Cynomolgus macaques at the early timepoint of assessment. Some differences in immune profiles were detected, with Cynomolgus macaques appearing to generate more robust IFN gamma responses.

Overall, no statistically significant differences were detected between the two species, however many of the end points had a small sample size. At a minimum, the authors should comment on the limitation of having a small sample size in the discussion.

We agree the sample size of two groups of 6 is relatively small and data from some of the latter endpoints cannot be subjected to statistical analyses. The animal experiment was carried out back in spring 2020 and little was known about the pathogenesis of SARS-CoV-2 infection in NHPs at the time. This animal experiment was key to develop the NHP model of infection at PHE Porton Down, which has been used successfully in the assessment of several vaccine candidates.

We have added in the discussion (line 522) “The sample size used in this study is relatively limited (6 rhesus and 6 cynomolgus macaques), and therefore, there are limitations to provide statistical significance for some endpoints. However, our results”

Major Comments:

1. A schematic with a timeline for various assessments would be helpful.

A new Figure (Fig. 1) has been added with a schematic to indicate the experiment and sampling timeline

2. For Figure 1, the text in the body of the manuscript indicates that 2 Cynos and 2 Rhesus were evaluated by CT. However, the Figure only shows 2 Cynos and 1 Rhesus. What is the basis of A and A'? How were these fields selected?

The images were selected to provide examples of the pulmonary abnormalities identified from CT scans collected from rhesus and cynomolgus macaques following infection with SARS-CoV-2. The figure has been refined to show the presence of peripheral ground glass opacities in all three macaques in which pulmonary abnormalities were identified in images A1, B1, C1, while images A2, B2, C2 show examples of either consolidation or other areas of ground glass opacification. Images from a second rhesus macaque (not shown) did not have abnormal features.

We have revised the CTscan figure (now Figure 2) and have added a more descriptive legend:

“Figure 2. Images constructed from CT scans collected 18 days after challenge with SARS-CoV-2 showing pulmonary abnormalities in two cynomolgus (A, B) and one rhesus macaque (C). Arrows in images A1, B1 and C1 indicate areas of peripheral ground glass opacification. Arrows in images A2 and C2 indicate areas of ground glass opacification and arrow in image B2 indicates an area of consolidation. Images from a second rhesus macaque did not have abnormal features”

What is the rationale for evaluating at 18 days post infection?

The CT scanner used for our studies is brought on site within a mobile unit when required and not permanently housed within our experimental facility. The study reported here was conducted during the first wave of the pandemic in the UK when the country was in lock down and consequently access to external services was restricted. However, CT scanning was incorporated into this study as soon as access to the CT scanner became possible.

Why were these particular animals selected for CT analysis? It would also have been helpful to include a CT from naïve animals for comparison.

To collect as much information as possible, CT scans were collected from all animals remaining in the study as soon as access to the CT scanner was possible.

The rationale for the figure was to provide examples of the pulmonary abnormalities observed after SARS-CoV-2 infection.

Our group at PHE has an historical collection of numerous CT scans from naïve rhesus and cynomolgus macaques, from our on-site closed breeding colonies, that have been analysed by the Consultant Radiologist reviewing the images collected in this study.

3. It appears that there is low level virus detected in the upper airways almost 3 weeks post challenge. Is this the most prolonged period of assessment in nonhuman primates?

Similar observations are made in a recent article has been made available as preprint (<https://doi.org/10.1101/2020.11.05.369413>) following animals up to to 21dpc, which showed virus load (PCR) up to 10 dpc.

Our group also found extended virus shedding for over 2 weeks in the ferret model (Ryan et al., Nat Comms in press).

4. The authors indicate that viable virus (as tested by plaque assay) could not be detected after 4 days post challenge, but viral RNA is detectable for 2 weeks following this period suggesting continued replication at this site. What could explain this discordance?

The qPCR used to detect viral RNA is orders of magnitude more sensitive than either plaque or TCID50 viable assays. Providing the possibility that virus replication may continue in the upper airway over this period resulting in overall quantities of shed viable virus that's are below the detection limit of the viable assay.

To evaluate this possibility further, we have performed sgRNA analysis of Nasal washes, throat swabs and BAL and have included these results in the new Figure 3. Moreover, we have performed RNA and sgRNA analysis in tissues: nasal cavity, lung, trachea and tonsil (new Supplementary Figure 1). We have not detected sgRNA from tissues at the second and third time points (14/15 and 18/19 respectively) and the sgRNA titres in the clinical samples are significantly lower than the total RNA.

In the light of these findings, it is likely that the low level viral RNA detected at later time points is not viable virus but killed viral material within the immune cells that are present in the airways.

5. Controls and associated images should be provided for in situ hybridization data.

We have added the following text "Appropriate positive and negative controls were included in each RNAscope run (Suppl. Figure 3)" in the Methods section and included a supplementary figure with images from negative and positive controls for the ISH runs (Supplementary Figure 3) as below:

Supplementary Figure 3. Negative and positive controls for in situ hybridisation techniques to detect virus RNA and IL6 mRNA. Negative control slides for ISH runs showing no staining in a rhesus macaque (A) and cynomolgus macaque (B) culled at 4 dpc. An irrelevant non-specific probe was used replacing the specific SARS-CoV-2 S-gene or IL-6 probes. Negative control naïve rhesus (C) or cynomolgus macaques (D) in ISH runs to detect SARS-CoV-2 S-gene. No staining was observed in naïve animals (archived samples from animals culled for non-infectious welfare reasons). Positive control slides to detect IL-6 mRNA using specific probes in rhesus (E) or cynomolgus macaques from a Mycobacterium tuberculosis infection experiment. IL-6 mRNA (red staining) is shown within granulomas.

6. There appears to be evidence of baseline neutralizing antibody in at least two of the Cynomolgus macaques. How did these animals in particular respond to infection? Some of the Cynomolgus macaques also appear to have much higher specific IgG responses for days 8-9. Are the highly responsive animals the same animals with elevated neutralizing antibodies at baseline? Is this evidence of a memory response (previous infection)?

There were 2 cynomolgus macaques with baseline levels of neutralising antibodies and the same two animals were among the highest responders at later time points.

The most-likely explanation for the baseline levels is potential cross-reactivity with common seasonal coronaviruses resulting from prior infection. There are recent reports about this cross-reactivity, e.g. Khan et al., <https://pubmed.ncbi.nlm.nih.gov/32511324/> using a peptide array based on common coronaviruses and reporting cross-reactivity with SARS-CoV-2.

7. Although it is understood that sample size is small, it would be of interest to comment on whether sex associated with any parameters in the study.

We agree with the reviewer the sample size is relatively small and we did not want to speculate about possible differences associated with sex. We found some evidence of potential sex differences in the transcriptomic analysis included in this reviewed version and included a sentence in the results section (line 423):

“There was some evidence of potential sex differences, although the N number maybe too low to draw specific conclusions (Suppl. Figure 4).”

We have added a sentence in the discussion about the small sample size and the limitations associated to provide statistical significance (lines 522-527):

“The sample size used in this study is relatively limited (6 rhesus and 6 cynomolgus macaques), and therefore, there are limitations to provide statistical significance for some endpoints. However, our results clearly demonstrate that both species provide infection models of SARS-CoV-2, that reflect upper and lower respiratory tract viral replication resulting in a lung injury, repair and resolution picture typical for milder forms of COVID-19 disease in humans”

Reviewer #2 (Remarks to the Author):

In this report, Salguero et al. compare SARS-CoV2 infection in rhesus macaques (RM) and cynomolgus macaques (CM) as animal models for studying COVID-19. This is important as there is a scarcity for the availability of rhesus macaques and the availability of additional species of non-human primates (NHPs) as an animal model will help to accelerate the development of immunotherapies and vaccines for COVID19. To accomplish the goals, the authors infected 6 CMs and 6 RMs with SARS-CoV-2 Victoria/01/2020 virus and performed viral RNA measurements and CT scans, antibody responses and T cell responses. Most of these analyses are thorough. The analyses revealed that both species show comparable levels of virological and immunological findings. The authors should address the following comments/concerns.

1) Fig. 1 needs proper legend.

We have revised the CTscan figure (now Figure 2) and have added a more descriptive legend:

Figure 2. Images constructed from CT scans collected 18 days after challenge with SARS-CoV-2 showing pulmonary abnormalities in two cynomolgus (A, B) and one rhesus macaque (C). Arrows in images A1, B1 and C1 indicate areas of peripheral ground glass opacification. Arrows in images A2 and C2 indicate areas of ground glass opacification and arrow in image B2 indicates an area of consolidation.

2) Identify groups in Fig 2.

We have included a new Figure 1 with a schematic of the experimental design. We included in the figure legend that results from rhesus macaques are in blue and cynomolgus macaques in red. We have included this information in the actual figure too.

3) The authors should write a sentence in the methods section about peptide pools covering which region of the protein. For example PP1 covers NTD of Spike protein, PP2 covers....!

We have included all the information from the peptide pools in a new Supplementary Table 1.

We have added the following information into the methods section “... and offset by 4 amino acids. Peptides were resuspended in 10% DMSO to a median concentration of 9.22 mg/ml. Peptides were then pooled and aliquoted. Before use peptides were resuspended to a concentration of 5 µg/peptide in media, they were then loaded onto the ELISpot plate, 50 µl/well. This peptide library is based on GenBank: MN908947.3 sequence. Full sequences of peptides are included in Supplementary Table 2.”

4) Provide a reference for the following sentence; Histopathological changes were comparable in both animal species, and closely resembled that seen in human cases following a mild/moderate clinical course (line 424).

We have added the following references in the discussion (line 468):

- **Tian et al. (2020). “Pulmonary pathology of early-phase 2019 novel coronavirus (COVID-19) pneumonia in two patients with lung cancer”. Journal of thoracic oncology (2020): DOI:<https://doi.org/10.1016/j.jtho.2020.02.010>**
- **Zeng et al (2020). “Pulmonary pathology of early-phase COVID-19 pneumonia in a patient with a benign lung lesion” Histopathology DOI: [10.1111/his.14138](https://doi.org/10.1111/his.14138)**

Reviewer #3 (Remarks to the Author):

Salguero and colleagues provide data on a comparative challenge experiment with SARS-CoV2 in Asian macaques where the primary variable would be the species of macaque used (rhesus vs cynomolgus). The study involves a small cohort (n=6 per species) of sex balanced, pubescent animals (2-4 years old, puberty in these species occurs between 2-5 years of age). After intranasal/IT challenge, the animals were monitored and sampled longitudinally by various methods. Scheduled euthanasia events were performed to track pathological changes in tissues on subsets of animals over time (n=2, sex balanced). The central claim of the work is that cynomolgus macaques (of Mauritian origin) challenged with SARS-CoV2 result in similar responses to challenge to that of rhesus macaques (Indian origin).

The manuscript needs a fair amount of work to clear up a number of serious issues.

1. Manuscript title: I would seriously blunt the claim of either of these models being “an authentic model of COVID19”. What is described here is an infection model at best.

We have changed the title and removed the word “authentic” as we agree not all the features observed in human COVID19 cases can be observed in the cynomolgus and rhesus macaque model. The new title reads “infection model” instead of “authentic model”

2. This reviewer agrees with the urgency to find alternative non-human primate models (NHP) that accurately recapitulate human COVID19 as the current trade blockade of this precious resource from Eastern Asia has no doubt put a serious strain on NHP resources across the globe. While the work holds some novelty as a limited comparison between two species and does demonstrate value of the Mauritian cynomolgus macaque as an infection model of SARS-CoV2, which much of the field would agree is comparable to what is known about rhesus as a model. The major caveat here is that cynomolgus macaques, regardless of origin, are almost as hard to source as rhesus currently. So authors claim that using cynomolgus as a replacement does not hold the perceived value claimed by the authors, at least in the eyes of this reviewer.

As we have stated in our manuscript, the availability of rhesus macaques is being compromised by the large amount of preclinical COVID-19 work that is so far focused on rhesus macaques. The urgent requirement for NHP COVID-19 studies in support of vaccine/therapeutic development will continue beyond the availability of this supply. Cynomolgus macaques are widely used for preclinical studies for a range of diseases that a small number of reports using them for COVID19 studies. There certainly is availability of significant numbers of this species from breeding colonies in Europe. Availability is only part of the value that we believe the cynomolgus macaque species can provide to the efforts to combat Sar-CoV2, as discussed in the manuscript the limited genetic diversity and the well characterised MHC of the Mauritian cynomolgus macaque population are key features that can be harnessed to advantage for studies.

This study serves to confirm that a transition to cynomolgus macaques, which to date remain available in multiple colonies, will make a significant contribution to the development and licensure of interventions against COVID-19.

3. A number of fundamental misconceptions are listed throughout the text that make it clear that the authors are not completely in tune with the state of the field. In particular, the authors assertion that a consensus that rhesus macaques are the “optimal study species” is unfounded (Abstract and lines 75-77). While it is true that early in the current pandemic a number of studies have used the model to assess vaccine candidates and development of immunity, these were largely done before more complete evaluations in other species were completed. The authors will no doubt realize that a number of investigators have independently validated the African Green Model as an equivalent, if not superior, model of human COVID19 (Hartman et al PLOS Pathogens 16(9): e1008903.

<https://doi.org/10.1371/journal.ppat.1008903>, Woolsey et al, bioRxiv 2020.05.17.100289; doi:

<https://doi.org/10.1101/2020.05.17.100289>, Blair et al, bioRxiv 2020.06.18.157933; doi:

<https://doi.org/10.1101/2020.06.18.157933> ; Cross et al., Virol J 17, 125 (2020).

<https://doi.org/10.1186/s12985-020-01396-w> , Johnston et al, bioRxiv 2020.06.26.174128; doi:

<https://doi.org/10.1101/2020.06.26.174128>). The authors also assert that for SARS-CoV is the preferred species are rhesus (LINE 66-69), which again is arguably false as the African Green Monkey model has clearly been demonstrated (in parallel) to be equivalent if not superior to both Asian

macaque species, especially rhesus (McAuliffe et al Virology. 2004 Dec 5; 330(1): 8–15, Smits et al., Journal of Virology Apr 2011, 85 (9) 4234-4245; DOI: 10.1128/JVI.02395-10 and other...). There is also growing understanding in the field that the rhesus is more of a viral clearance model, than one truly reflective of global pathogenesis reflective of human COVID19. While infection models are certainly valuable for demonstrating immune responses, without demonstration of associated pathology, their value is limited in terms of detailed pathological studies or even evaluation of predictive efficacy of therapeutics (especially post exposure).

We agree there are several studies using African Green monkeys as a model for COVID19, and it has been demonstrated to be a very valuable model. Our assertion about the accepted preference of rhesus macaques comes from the amount of publications using the rhesus model in vaccine and therapeutic trials. Even though there are publications about model development in different NHP species, the far majority of published therapeutic preclinical trials to date have been carried out in rhesus macaques. This fact has been discussed in different meetings and we have included an independent paper about it in the reference list: Zhang, S. America Is Running Low on a Crucial Resource for COVID-19 Vaccines. The Atlantic (2020).

Lu et al published in Nature Signal Transduction and Targeted Therapy (<https://www.nature.com/articles/s41392-020-00269-6>) a study comparing old world and new world monkeys to identify a suitable model for COVID-19 stating that "... *M mulatta* as the most suitable model for modelling COVID-19" in contrast with cynomolgus macaques and marmosets.

We have included this reference in the introduction and discussion and have deleted this sentence from the introduction "There is an accepted preference to use rhesus macaques for the assessment of COVID-19 vaccine and therapeutics, based on the limited number of studies that have been performed".

One of the main conclusions from our study is that Mauritian cynomolgus provide a model of SARS-CoV-2 infection that is analogous to the model provided by rhesus macaques, and therefore this provides an alternative source of subjects for study to alleviate the pressure on the dwindling stocks of rhesus macaques. Use of the Mauritian cynomolgus macaque can also bring some other attributes that could be exploited as discussed above. We are not stating they are the best model, but an alternative with similar characteristics. Mauritian cynomolgus macaques are not Asian and originated from a small founder population brought to Mauritius by trading ships hence the limited genetic diversity which makes this population a useful scientific tool.

We are not aiming to compare the strengths and weaknesses of every NHP species as COVID19 models.

4. BAL collection: This is not a standard way of collecting BALs as reported in all other NHP SARS-CoV2 studies, this is more of an ex-vivo flushing of the lung lobe(s?). Given the animals are euthanized via intracardiac injection, there is major suspect to the validity of any data gleaned from these samples due to potential artifact associated with cellular contact with the euthanasia solution. The data for RNA is likely good, but I'm not sure I would be confident in any cellular immunological or histological analysis done on lungs sampled this way.

We have collected the BAL fluid at post-mortem. We have a robust systematic way of collecting this sample for downstream analysis using the right lung, ligated and separated from the left lung. The latter was sent intact for histopathological analysis. With this protocol, we avoid any possible induced artefact in histopathology. We have an extensive track record working with NHPs and we are confident no artefacts are induced by the intracardiac injection.

5. Nasal washes: Why not just swab? This is what the majority of the field is doing and is more reflective of human sampling. Nasal washes are typically done on smaller animal models such as ferrets or hamsters as swabbing is not practical approach. The approach makes comparability with other studies challenging.

We performed nasal wash instead of swabs to get obtain more cells for downstream RNAseq analysis, the resulting transcriptional analysis are now included in this updated manuscript.

6. Line 151: This is high? Was virus isolated from these samples? If high, there should have been no problem. Between 6-7 logs of virus is just about the threshold of where one could expect to have any success isolating infectious virus from a biological sample in such a way that it could be effectively quantified by plaque assay. I would temper the claim.

We have carried out Subgenomic RNA analysis, indicative of replicating virus, in *in life* samples as well as in tissue samples from the post-mortem. New figure 3 and new supplementary figure 1 include the expanded genomic and subgenomic PCR data.

Subgenomic RNA detected 1.7×10^5 copies/ml (sgRNA) at 5 dpc for rhesus and 4.8×10^4 copies/ml (sgRNA) at 4 dpc for cynomolgus macaques (Fig. 3F).

7. Line 170 and throughout. Virus was only isolated (at very low levels) 1-3 dpc, how do you know this was not just left over inoculum from the challenge?

We have analysed the subgenomic PCR as to further evaluate replicating virus (new figures 3 and supplementary Figure 1).

8. Line 386, not the first time: Johnston et al, bioRxiv 2020.06.26.174128; doi: <https://doi.org/10.1101/2020.06.26.174128>

We have deleted this claim in line 386 (discussion) and in the introduction (line 84)

9. There is mention of credentials of the radiologist as being an "expert". What does that mean? Related to this, are the pathologists equally qualified? This would be important justification for trying to develop a "new method for lung histopathology scoring". Abstract, line 42.

The radiologist (co-author) is a medical radiologist expert in respiratory diseases (30 years' experience of interpreting Chest CT scans (FG) including 10 years' experience with macaque

CT scans). He has been working on COVID19 radiology/imaging in a reference hospital since the start of the pandemic.

There are four pathologists as co-authors, three qualified veterinary pathologists (FJS, ELR and GP) and one qualified medical pathologist (AGN) (all Fellows of the Royal College of Pathologists).

We have added this information in the Methods sections: (line 621) Sections were analysed by two qualified veterinary pathologists and peer-reviewed by two other qualified pathologists (veterinary and medical respectively).

10. Study terminology: The description is somewhat confusing, what does 4/5 and 14/15 etc mean? Did the authors take body fluids on the first day (4) and then culled on the second day (5)? Introduction, lines 91-93.

We agree this can be confusing. The animals were culled in pairs at 4 or 5 dpc, etc. We have included a new Figure 1 with a schematic of the study to make this clear.

11. Why was CT only done on limited animals and only on day 18, why not before when the virus is running rampant in the lungs? With no kinetics of changes over time, these data hold little value. Results, lines 114-115.

The CT scanner used for our studies is brought on site within a mobile unit when required and not permanently housed within our experimental facility. The study reported here was conducted during the first wave of the pandemic in the UK when the country was in lock down and consequently access to external services was restricted. However, CT scanning was incorporated into this study as soon as access to the CT scanner became possible.

12. The only thing noted on CT was ground glass opacity but later in the text the authors mention they say that "Features characteristic of COVID-19 in human patients, such as ground glass opacity, consolidation and crazy paving were identified on CT scans collected from both SARS-CoV-2 challenged rhesus and cynomolgus macaques in line with...." Discussion, line 401-404 vs Results, lines 120-125. Reconciliation needed.

We describe in the results that ground-glass opacity was observed in the three macaques showing lung abnormalities with peripheral consolidation in one cynomolgus macaque. We have deleted "crazy paving" from the sentence in the discussion

13. "Gross pathological changes were found in the lungs of all animals....", so where are the gross images? How about a gross score to determine severity and to correlate with the histopath scoring?....Pathological changes, lines 176-181.

We have included gross pathology images from one cynomolgus and one rhesus in a supplementary figure (Supplementary Figure 1). The gross pathology observed was multifocal and mild, and we did not develop or apply any scoring system.

14. In regards to tissue handling, were the lungs sufflated? were all lung lobes sampled? I don't think so because the M&M stats this... "...BAL was collected at necropsy from the right lung. The left lung was dissected prior to BAL collection and used for subsequent histopathology and virology procedures..." I think there are inherent errors if not all lung lobes are collected and evaluated for this disease since it's so multifocal and mild so if trying to develop a method for scoring we should only look at half the lung? Also if lobes are not sufflated it is really difficult from animal to animal to determine alveolar thickness due to inflammatory infiltrates vs just collapse with multiple alveolar walls side by side. Post-mortem examination and histopathology, lines 610-614.

Lungs were not insufflated. The left lung lobes were ligated and removed during the post-mortem procedure. BAL fluid was then obtained from the right lung lobes. The left cranial and caudal lobes were fixed by immersion in buffered formalin, after taking small samples for virology and molecular biology. With our experimental procedure, it was not possible to sample from all lobes for every test performed. However, multiple sections from the left lung lobes, spanning proximal and distal locations, were evaluated in detail to account for the multifocal nature of this disease, and this was considered sufficient to provide us with an overall estimation of pathology within this organ.

We developed the scoring system for microscopic evaluation of changes to evaluate individual slides whilst being blinded to both the species and dpc. Even though the lesions are multifocal and mild, we have been able to successfully apply this scoring system to subsequent studies (preclinical vaccine trials) and have observed a significant correlation between the CT scan scores (of the whole lung) and the histopathology scores (left lung lobes only).

We appreciate that not insufflating the lungs may limit the evaluation of subtle changes in alveolar walls and this was considered carefully in the experimental design procedure. However, the insufflation process also induces other type of artefacts. With our combined expertise in both NHP respiratory pathology and human pathology, we were confident that the absence of insufflation would not significantly limit our overall interpretation of lung-induced changes due to this infection.

15. Along the lines of #8, how many sections of lung were examined and from where? Most seasoned pathologists working with respiratory viruses make a big deal about sampling the hilar portion and the distal portion. 3 per lobe would be considered standard practice. It is stated that the authors take 3 samples, but they do not define where the samples are taken from... "Three tissues sections from each left lung lobe were used to evaluate the lung histopathology." Also did they only define the cranial and caudal portions? or did they take the cranial (upper lobe) and caudal portions (middle lobe) of the cranial lobe and the caudal lobe (Lower lobe) so we are not talking looking at 6 slides vs 9 slides vs the 18 slides that we look at, there always is variability when slides are reviewed, so a more clear understanding of the pathologists process in determining confidence in thier scoring would be particularly helpful to build confidence in the proposed scoring method. Post-mortem examination and histopathology, lines 625-626.

We sampled 6 sections of the left lung, three from each lobe; all spanned the entire lobe. In these species the left lung is divided into two distinct lobes. We have used the most common veterinary nomenclature for the lobe names: cranial and caudal, instead of upper and lower, which is considered to be clear and appropriate.

We have expanded the description of the lung samples sectioning in the text. "Six large tissue sections from the left lung were used to evaluate the lung histopathology, three from the cranial lobe and three from the caudal lobe as follows: a) proximal to the lobar bronchus bifurcation, b) middle portion and c) distal to the lobar bronchus bifurcation." (line 667-)

The scoring system was developed and refined by consensus of the pathologists involved in the project.

16. There is a lot of heavy use buzz word of DAD. There are some specific requirements for you to hang your hat on the diagnosis of DAD, and the data as provided don't seem to back this up. DAD means diffuse alveolar damage. It's hugely important in human COVID-19 but the lesions really are not that impressive in all of the animal models (with the exception of the Mink--there is some beautiful lesions in mink). The first thing that qualifies DAD is the presence of hyaline membranes, this paper fails to point out any hyaline membranes by IHC. Its likely however, that the authors may be challenged to observe this feature without proper sufflation of the lungs during tissue preparation. I think the authors could conclude there was alveolar damage but not DAD unless they provide the hyaline membranes and the severity of the lesions (aka it needs to be diffuse (the whole lung) otherwise it's focally diffuse. "Overall, diffuse alveolar damage (DAD) was a prominent feature in the affected areas (see it's focally), characterized by individual, shrunken, eosinophilic cells in alveolar walls with pyknotic or karyorrhectic nuclei" Pathological changes, lines 189-191.

We agree the term DAD is used in medical pathology when several features are present, including the formation of hyaline membranes, which are not present in the NHP model we are describing here. However, we can observe, a multifocal damage of the alveoli, primarily as necrosis of the alveolar epithelial lining, which, in itself, defines damage to the alveoli. To avoid confusion, we have removed DAD from the description and have described the alveolar necrosis and associated lesions accordingly. The presence of hyaline membranes can be observed in human cases of COVID. We have worked with tissue sections from several human patients that died of COVID19 (fatal cases) and many of them showed hyaline membranes. The human samples were not insufflated and even though the autolytic changes are more evident in the human cases, and no insufflation was performed, the hyaline membranes were easily identifiable.

17. More path issues to address...

a. "in these areas, alveolar spaces were often obliterated by collapse of the thickened and damaged alveolar walls..." If the lungs were not sufflated they are expected to collapse when the air escapes at opening the chest cavity/removal of the pluck. Pathological changes, line 192.

We acknowledge that artefacts can result due to the loss of pressure that occurs when opening the thoracic cavity, and we are competent at routine identification of such changes. However, what we are describing here are distinct from these, comprising thickened and damaged alveolar

walls that can be easily identified, with particular prominence during the first time point post-challenge, and there is a good correlation with the presence of large quantities of virus RNA.

b. "...admixed with fibrin, polymorph neutrophils (PMNs), enlarged alveolar macrophages and other round cells (possibly detached type 2 pneumocytes)." Ok, PMN stands for polymorphonuclear cells not polymorph neutrophil. also PMN includes all granulocytes so, so may be best to just say neutrophils here; Pathological changes, lines 197-198.

We have changed "polymorph neutrophils (PMNs)" to "neutrophils"

c. "enlarged alveolar macrophages" how do you know they are enlarged? Typically they are described as activated or reactive i don't think it's proper to say enlarged (although it does get the point across) and if either of these its suggested to demonstrate activation or reactivity with a specific marker. Pathological changes, lines 197-198.

We have described alveolar macrophages as "enlarged" as we believe this is the most appropriate morphological description. Our confidence in describing such changes arises from the ability for direct comparison of the dimensions of macrophages within the more "healthy" parenchyma within the same or different tissue sections. Furthermore, we have extensive, collective experience in the assessment of pulmonary pathology in numerous NHP animal experiments. and we can also compare with the macrophages in the "healthy" areas of the lung even within the same tissue section. We agree these cells are likely to be activated; however, as we have not performed any special technique to verify this, the morphological changes are simply being described appropriately.

d. "other rounds cells" this means leukocytes like lymphocytes, plasma cells etc...a pneumocyte is not a round cells it's epithelial and you can easily run an IHC marker to determine (as many others in the field have) so consider running a pancytokeratin or surfactant IHC to be sure and not to guess. Also I don't think it's proper to say type 2 it's always noted as type II in the roman numbers. Pathological changes, lines 197-198.

We have changed the text to "enlarged alveolar macrophages, few lymphocytes and detached type II pneumocytes"

e. "In distal bronchioles and bronchiolo-alveolar junctions, degeneration and sloughing of epithelial cells was present, with areas of attenuation and foci of plump, type 2 pneumocytes representing regeneration." This could also be bronchiolization so the epithelium is coming from the bronchiolar epithelium and not just the type II pneumocytes that are in the alveolar walls. Need to define where you see this. Type II pneumocytes hyperplasia will likely not be in the distal bronchioles but more likely in the alveolar junctions. Pathological changes, lines 198-201.

We have changed the sentence to "In distal bronchioles, degeneration and sloughing of epithelial cells was present, with areas of attenuation; in alveolar walls associated with bronchiolo-alveolar junctions, foci of plump, type II pneumocytes were noted, representing regeneration".

f. PMN issue again, just need to put neutrophils. Continuing to describe these as PMN confuses the issue with generalized granulocytes. Also, is this inflammation associated with the previously

described epithelial sloughing? otherwise the sloughing maybe a postmortem change and not a pathologic change. Pathological changes, line 206.

We have changed “PMNs” for “neutrophils”. The sloughing can be a post-mortem change, but we have clearly described this change in association with inflammation and regeneration. Inflammatory changes are not present in the artefactual, post-mortem changes.

g. "In the lumen of some airways, fibrillar, eosinophilic material (mucus)...." Mucus is always slightly basophilic or amphophilic...eosinophilic is edema or fibrin. Also mucus is not fibrillar that is really pointing towards fibrin...so its suggested to run a fibrin stain (PTAH or IHC). Pathological changes, lines 207-208.

We agree mucus is basophilic. This was an oversight and so have changed “ fibrillar, eosinophilic material (mucus) for just “mucus”.

h. "...resembling syncytial cells..." please don't guess, run IHC for epithelial cells or macrophage marker or just leave it at multinucleated cells. Pathological changes, lines 210-211.

We removed the speculation about “resembling syncytial cells”, and left just “multinucleated cells”

i. "...and hyperplasia of alveolar macrophages..." this is a strange term, standard terminology is to say increased numbers of alveolar macrophages because they are recruited there rather than generated there. Pathological changes, line 217.

We have changed “hyperplasia” for “accumulations”

j. "Positive cells were also observed rarely..." What cells are they, mononuclear cells? Pathological changes, line 225.

We have added “mononuclear cells”

k. "...expressing IL-6 mRNA..." Not an overly convincing way to say that IL-6 was associated with the lung tissue inflammation in any meaningful way without evidence at the protein level and perhaps more convincing association with inflammation and viral antigen (or even RNA). Also no cytokine analysis at the protein level to support actual generation of the marker rather than a few cells lighting up with the marker? Transcription does not always equal translation... Pathological changes, line 227 also listed in Discussion, lines 435-439.

We have observed numerous positive cells to IL-6 mRNA using our ISH hybridisation technique, and these are in prominent, intralesional locations. We agree we do not show evidence of an increase of IL-6 at the protein level, but we are not claiming that fact. It is believed that the upregulation of IL-6 at the transcript level is at least sufficient to discuss the observed elevation of IL-6 levels in human patients. We have changed “interleukin-6 production” for “interleukin-6 transcription at the mRNA level”

l. "...rhesus macaques within the alveolar walls and interalveolar septa..." what's the difference?...alveolar walls and interalveolar septa? Unless the authors mean pneumocytes (as the alveolar walls) and capillaries with macrophages (as the interalveolar septa)? This seems very poorly defined and unfounded. Will require better descriptions Pathological changes, line 231.

We have reformatted the whole ISH section. The first paragraph describing the presence of SARS-CoV-2 reads now as (line 254-):

"Presence of viral RNA was observed in the lungs from all animals at 4/5 dpc by in situ hybridisation. Prominent staining of small foci of cells containing SARS-CoV-2 viral RNA, was observed within pneumocytes and interalveolar septa, concomitant with microscopic changes in cynomolgus macaques (Figure 6A). Staining was not seen in cells or fluid within the alveolar spaces. Positive mononuclear cells were also observed rarely in the bronchus-associated lymphoid tissue (BALT) (Figure 6A, insert). Small foci of cells staining positive for viral RNA were observed at a low frequency in the rhesus macaques within the alveolar lining and interalveolar septa of both animals at 4/5 dpc, concomitant with microscopic changes (Figure 6B)."

18. "In summary, using the histopathology scoring system developed here, the scores..." Where are the definitions of the scoring done? As provided, this reviewer cannot get a clear view understanding that the paper was written using the criteria in the table provided. I think at best that can be said here is how they graded the left lung lobes when comparing rhesus and cynos in this paper. I don't think this will be a new standard of scoring, also any pathologist will be looking at these areas and i think it's only applicable to the early stages of the disease, what about fibrosis later in the disease? As no hyaline membranes represented as part of the scoring, the value of the new method is further reduced as this feature is extremely important in humans and is good to know if they are present in animal models. Also do they score this per animal or per lung lobe or per slide examined? What is done if there is not a bronchus in your section? How do you include IHC or ISH scoring? The scoring is on a percent basis, so how do you assess that? Look at 10 high power fields and average or you count the whole slide? What if the tissue sample is minimal? Lots of questions in regards to the robustness of this scoring system but do think these are valid lesions to look at and consider when reading out the slides, just not sure a score is needed for each of these. Once you get a score how does that correlate to clinical or blood pathologies? Pathological changes, line 251.

The development of a new scoring system in histopathology is a complicated ask. We have been reading many slides from different NHP SARS-CoV-2 studies before producing the final version of the scoring system used here. One of the first approaches was adapting, with significant changes, a previously described scoring system developed for influenza virus infection in pigs as a model for human influenza (Morgan et al., 2016: <https://www.ncbi.nlm.nih.gov/pmc/articles/PMC4891568/>) that we have used previously in our group. This system also uses percentages for several parameters. Using the percentages, we have been able to reduce subjectivity and decrease dramatically the discrepancies among pathologists reading the slides. We use digital scanning and are able to analyse the entire slides instead of going through different high-power fields. As we use large sections from different lung lobes, we do not have the problem of getting minimal tissue samples. We discussed the possibility of not having bronchi in a slide. The scores are the averages

of the individual scores for each slide. In case we do not see bronchi in one slide, we still have the scores from the other slides from the same animal.

We have not included parameters that we do not observe in our model, like hyaline membranes or fibrosis and that have not been consistently described in other NHP SARS-CoV-2 studies.

Using only the left lung lobes may not reflect what pathology is happening in the rest of the lung. We have used successfully this score system in at least 4 further studies with rhesus macaques with very good correlation with PCR data from the same lung lobes, and CT scan data from the entire lung.

Regarding ISH/IHC, we do not include the results in the scoring system, and we have developed a specific way to score the presence of virus RNA and IL-6 mRNA using digital image analysis, providing objective results.

13) Refine the summary paragraphs, I think just needs to be reworded to tie everything together, seems a little thrown together. Pathological changes, lines 251-273.

We have reorganised the section of pathological changes, starting with the histopathology (H&E), and followed by the ISH results, which follows a similar format to that used for reorganising the figures (see comments at the bottom - Figure legends).

The pathology section reads now as:

Gross pathological changes were found in the lungs of all animals from both species and sexes euthanised 4/5 days after challenge, and comprised multiple areas of mild to moderate consolidation, distributed in cranial and caudal lobes (Suppl. Figure 1). At 14/15 and 18/19 dpc, only small areas of consolidation were observed. Macroscopic changes, considered to be associated with infection, were not observed in any other organ analysed in this study at any time point.

Histological changes in the lungs of all twelve animals from both species, consistent with infection with SARS-CoV-2, were observed. The changes were most prominent at 4/5 dpc and thereafter were less severe, indicating resolution of the more acute changes observed at early time points.

4/5 days after challenge, the lung parenchyma in the cynomolgus macaques was comprised of multifocal to coalescing areas of pneumonia, surrounded by unaffected parenchyma. Overall, alveolar necrosis was a prominent feature in the affected areas, characterised by individual, shrunken, eosinophilic cells in alveolar walls, with pyknotic or karyorrhectic nuclei (Figure 4A). In these areas, alveolar spaces were often obliterated by collapse of the thickened and damaged alveolar walls which contained mixed inflammatory cells (Figure 4A,B); or had obvious, alveolar type II pneumocyte hyperplasia (alveolar epithelialisation), as well as expanded alveolar spaces (Figure 4B). Alveolar spaces were expanded and filled with fibrillar to homogenous, eosinophilic, proteinaceous fluid (alveolar oedema) (Figure 3A), admixed with fibrin, neutrophils, enlarged alveolar macrophages, few lymphocytes and detached type II pneumocytes. In distal bronchioles, degeneration and sloughing of epithelial cells was present, with areas of attenuation; in alveolar walls associated with bronchiolo-alveolar junctions, foci of plump, type II pneumocytes were noted, representing regeneration. In the larger airways, occasional, focal, epithelial degeneration and sloughing was observed in the bronchial epithelium, with evidence of regeneration

characterised by small, basophilic epithelial cells. Low numbers of mixed inflammatory cells, comprising neutrophils, lymphoid cells, and occasional eosinophils, infiltrated bronchial and bronchiolar walls. In the lumen of some airways, mucus, admixed with degenerative cells, mainly neutrophils and epithelial cells, was seen. Occasionally, multinucleated cells, characterised as large, irregularly shaped cells with prominent, eosinophilic cytoplasm and multiple round nuclei, were observed (Figure 4B, inset).

Pathological changes consistent with those described for cynomolgus macaques were present in the lungs of rhesus macaques. In the parenchyma, multifocal expansion and infiltration of alveolar walls by inflammatory cells was noted (Figure 4C). Furthermore, in these areas, alveolar necrosis was observed with patchy alveolar oedema and accumulations of alveolar macrophages (Figure 4C). In the bronchi and bronchioles, similar changes to those described for cynomolgus macaques were seen (Figure 4D).

Changes were less severe in all four animals examined at 14/15 dpc. In the cynomolgus macaques, patchy infiltration of mainly mononuclear cells in the alveolar walls, with occasional similar cells within alveolar spaces, and parenchymal collapse, were seen (Figure 4E). Mononuclear cells, primarily lymphocytes, were also noted surrounding and infiltrating the walls of blood vessels (Figure 4F) and airways. Furthermore, there was an increased prominence of bronchial-associated lymphoid tissue (BALT). In the lungs of rhesus macaques, changes in the alveoli and BALT were similar in appearance and frequency to those described in the cynomolgus macaques, and perivascular lymphocytic cuffing of small vessels, characterised by concentric infiltrates of mononuclear cells, was also seen occasionally (Figure 4G and 4H). By day 18/19, the changes were similar but less frequent to those described at day 14/15 in all four animals.

Using the subjective histopathology scoring system, scores were higher in both macaque species at 4/5 dpc compared to 14/15 and 18/19dpc, mostly due to higher scores in the alveolar damage parameters observed at the early time point (Figure 5).

Presence of viral RNA was observed in the lungs from all animals at 4/5 dpc by *in situ* hybridisation. Prominent staining of small foci of cells containing SARS-CoV-2 viral RNA, was observed within pneumocytes and interalveolar septa, concomitant with microscopic changes in cynomolgus macaques (Figure 6A). Staining was not seen in cells or fluid within the alveolar spaces. Positive mononuclear cells were also observed rarely in the bronchus-associated lymphoid tissue (BALT) (Figure 6A, inset). Small foci of cells staining positive for viral RNA were observed at a low frequency in the rhesus macaques within the alveolar lining and interalveolar septa of both animals at 4/5 dpc, concomitant with microscopic changes (Figure 6B).

Viral RNA was detected in only a few individual cells in both groups of animals at 14/15 dpc (Figure 6C and 6D). By day 18/19, viral RNA was not detected by ISH. Overall, there was a high presence of viral RNA at 4/5 dpc which was more pronounced in the cynomolgus macaques; by contrast, only very, few positive cells were observed at 14/15 dpc and none at 18/19 dpc (Figure 6E).

Viral RNA was observed in scattered epithelial cells in areas of the upper respiratory tract (nasal cavity, larynx and trachea) of all animals at 4/5 dpc, and not associated with lesions.

Abundant numbers of cells expressing IL-6 mRNA were observed within the pulmonary lesions (Figure 3D), with only few positive scattered cells in the healthy parenchyma. IL-6 mRNA was also

abundant within the lesions in cynomolgus (Figure 6F) and rhesus macaques (Figure 6G) at 4/5 dpc, which was slightly more pronounced in cynomolgus macaques (Figure 6J). The expression of IL-6 mRNA was less pronounced in both cynomolgus (Figure 6H) and rhesus macaques (Figure 6I) at 14/15 dpc, with no significant expression observed at 18/19 dpc.

In the liver, microvesicular, centrilobular vacuolation, consistent with glycogen, together with, small, random, foci of lymphoplasmacytic cell infiltration were noted rarely (data not shown). This is considered to represent a mild, frequently observed, background lesion. Remarkable changes were not observed in any other tissue. Viral RNA staining was seen only at 4/5 dpc, in occasional, epithelial and goblet cells in the small and large intestine. It was not observed in any other tissue examined.

14) I don't know what lung MNC samples are, i either missed it or it's not defined. Composition and functional profile of the cell mediated immune response, line 325.

We have included the definition of MNC: “mononuclear cells“

15) "...mild/moderate clinical course. Typical changes of ARDS were observed, with ..." ARDS is not a mild/moderate clinical course (that i'm aware of, am i wrong?) I think they can say something like focal (diffuse) alveolar damage, pneumonia, type II pneumocyte hyperplasia, etc were noted...additionally they go on to say, "these histopathological lesions are compatible to those observed in human patients..." they aren't because nobody is getting biopsies (Tom and I had to search hard for non fatal histology..there's like 2 papers out there)...the human publications are largely only fatal cases and the NHPs are not representative of that.....Discussion, line 424-428

We have deleted ARDS as this is a medical term not compatible with some of the veterinary nomenclature. We have change ARDS to “acute respiratory viral infection”

We agree there are just a handful of papers out describing the milder forms of COVID-19 in human patients as biopsies are not being taken. However, they describe similar features to those observed in our NHP model

We have added the following references in the discussion:

- Tian et al. (2020). “Pulmonary pathology of early-phase 2019 novel coronavirus (COVID-19) pneumonia in two patients with lung cancer”. *Journal of thoracic oncology* (2020): DOI:<https://doi.org/10.1016/j.jtho.2020.02.010>
- Zeng et al (2020). “Pulmonary pathology of early-phase COVID-19 pneumonia in a patient with a benign lung lesion” *Histopathology* DOI: [10.1111/his.14138](https://doi.org/10.1111/his.14138)

16. Only two animals per timepoint per species were sampled. The animal to animal variability is very high with this virus, there is no way to conclude statistical significance on such a limited number of samples.

We agree the individual variability is high in this infection and subsequently, we have not made any claims regarding statistical significance. The animal experiment, with serial post-mortem time

points, was carried out at the early stages of the pandemic and was approved by a rigorous Ethical Review process.

FIGURE LEGENDS

Figure 1: The CT is not very sharp, I also don't know what A vs A' is, Is this a positional demarcation?

We have changed the figure legend (now Figure 2 as Figure 1 is new – schematic of the experiment)

Figure 2. Images constructed from CT scans collected 18 days after challenge with SARS-CoV-2 showing pulmonary abnormalities in two cynomolgus (A, B) and one rhesus macaque (C). Arrows in images A1, B1 and C1 indicate areas of peripheral ground glass opacification. Arrows in images A2 and C2 indicate areas of ground glass opacification and arrow in image B2 indicates an area of consolidation.

Figure 3: B: No hyaline membranes shown with the DAD, If it's true DAD there should be some hyaline membranes.... should be type II pneumocyte hyperplasia not just pneumocyte hyperplasia and again please run IHC for epithelial cells, don't guess for the multinucleated cells

D: DAD? Ok but again no hyaline membranes

F: alveolar macrophage hyperplasia? that's a strange term it's increased alveolar macrophages.

G: bronchial exudates? I don't see this here in the image.

I: parenchymal collapse (it should if you didn't sufflate the lungs)

J: perivascular cuffing? I can't tell if this is a vessel or not need closer image also more of a cuff if possible

K: bronchiole regeneration? I'm not sure why they say this here i can't see other than epithelium lining a bronchiole so maybe it's piling up?

We have made significant changes in the figures 3 and 4 and have expanded them into the new figures 4, 5 and 6.

The histopathological images (H&E staining) are in the new figure 4 with the new figure legend:

New figure 4. Histopathological changes in cynomolgus and rhesus macaques during SARS-CoV-2 infection. Diffuse areas of alveolar necrosis observed in cynomolgus macaques at 4/5 dpc with shrunken, eosinophilic cells within the alveolar walls (A, B), together with alveolar oedema (A, arrows), type II pneumocyte hyperplasia and expanded alveolar spaces with inflammatory cell infiltration (B, arrows). Occasional multinucleated cells are observed (B, insert). Similar histopathological changes observed in rhesus macaques, including alveolar necrosis and areas with patchy alveolar oedema (C, arrow), and accumulation of alveolar macrophages (D, arrow) and bronchial exudates (D, insert). Histopathological changes with less severity observed at 14/15 dpc in cynomolgus macaques, with infiltration of mononuclear cells within alveolar spaces and bronchiolar lumen (E, arrows) and perivascular cuffing (F, arrow). Bronchiole regeneration (G, arrow) and perivascular/peribronchiolar cuffing observed in rhesus macaques at 14/15 dpc (H, arrows), together with BALT proliferation (H, *).

We have removed the term DAD and we have used “alveolar necrosis” instead.

We have changed the term, alveolar macrophages hyperplasia for “alveolar macrophage accumulations”.

We have included an image of the bronchial exudates (D, insert).

We have removed the term parenchymal collapse.

We disagree with the comment about perivascular cuffing. We have included images in both the rhesus and the cynomolgus macaques showing blood vessels with mild to moderate perivascular cuffing (new Figure 4 F and H). The cuffing is not marked/severe, as reflected in the scores for these animals at the second time point.

We have kept the image for bronchiolar regeneration (Figure 4 G). Obviously not all the images can be shown in a full page and it may be difficult to interpret when the panel images are reduced, but we are confident with our description.

We have not included any image of the third time point to avoid repetition, as the manuscript has already a large number of graphs and panels.

We have removed the images of viral RNA and IL-6 mRNA and have integrated them in the new figure 6.

Figure 4: Confusing figure. Would like to just see the histology images of the lesions and the lack of lesions at the later time points. I don't really see the big difference in the histology, maybe it need to be presented differently?

We have redone the histopathology and ISH figures to avoid confusion. We have included Figure 4 with only histopathological images (H&E staining).

Figure 5 shows the heatmap with the individual scores for each animal at each time point. We think this is an important way of showing the histopathological, results and have been using it for several subsequent studies carried out with NHPs by our group.

Figure 6 shows the ISH results (images and graphs) for virus RNA and IL-6.

Reviewers' Comments:

Reviewer #1:

Remarks to the Author:

Authors have appropriately responded to reviewer comments.

Reviewer #3:

Remarks to the Author:

The authors have adequately addressed all of my concerns or at least provided appropriate rationale to justify the approaches and claims made. All of the clarifications were much appreciated by this reviewer as I am aware it was no small task. I would further argue the field will also appreciate the new data, clarity and the transparency provided by the changes made.